# Application of machine learning algorithms to identify serological predictors of COVID-19 severity and outcomes

Santosh Dhakal[1,11,12], Anna Yin[1,12], Marta Escarra-Senmarti [2,12], Zoe O. Demko[3], Nora Pisanic[4], Trevor S. Johnston [4], Maria Isabel Trejo-Zambrano[2], Kate Kruczynski[4], John S. Lee[1], Justin P. Hardick[5], Patrick Shea[1], Janna R. Shapiro[1], Han-Sol Park[1], Maclaine A. Parish[1], Christopher Caputo[1], Abhinaya Ganesan[1], Sarika K. Mullapudi [3], Stephen J. Gould[6], Michael J. Betenbaugh[7], Andrew Pekosz [1,4,5], Christopher D. Heaney [4,8,9], Annukka A. R. Antar [3], Yukari C. Manabe [1,3,9], Andrea L. Cox [1,5], Andrew H. Karaba [3], Felipe Andrade [2], Scott L. Zeger[10] & Sabra L. Klein [1,5,9] ✉

## Abstract

**Background** Critically ill hospitalized patients with COVID-19 have greater antibody titers than those with mild to moderate illness, but their association with recovery or death from COVID-19 has not been characterized.

**Methods** In a cohort study of 178 COVID-19 patients, 73 non-hospitalized and 105 hospitalized patients, mucosal swabs and plasma samples were collected at hospital enrollment and up to 3 months post-enrollment (MPE) to measure virus RNA, cytokines/ chemokines, binding antibodies, ACE2 binding inhibition, and Fc effector antibody responses against SARS-CoV-2. The association of demographic variables and more than 20 serological antibody measures with intubation or death due to COVID-19 was determined using machine learning algorithms.

**Results** Predictive models reveal that IgG binding and ACE2 binding inhibition responses at 1 MPE are positively and anti-Spike antibody-mediated complement activation at enrollment is negatively associated with an increased probability of intubation or death from COVID-19 within 3 MPE.

**Conclusions** At enrollment, serological antibody measures are more predictive than demographic variables of subsequent intubation or death among hospitalized COVID-19 patients.

## Plain language summary

Part of the adaptive immune response to viruses, such as SARS-CoV-2, is production of antibodies that are specific to the virus. Hospitalized patients with severe COVID-19 produce more antibodies against SARS-CoV-2 than patients with mild to moderate disease. We studied antibody responses in people with COVID-19 until either recovery or death from the disease. Among hospitalized patients, we analyzed factors, including demographic characteristics, comorbidities, and antibody features that could be used to predict the requirement of intubation or the occurrence of death from COVID-19. We found that antibody measurements taken when people were admitted to the hospital were better at predicting adverse COVID-19 outcomes than either demographic characteristics or comorbidities. These predictive measurements could be useful indicators of disease severity during future pandemics.

Most SARS-CoV-2 infections cause mild to moderate disease and do not require hospitalization[1]. Severe disease (i.e., hospitalization or intensive care unit (ICU) admission) and fatal outcomes are associated with older age, male sex, underlying comorbidities, and lack of vaccination[2,3]. Antibodies protect against SARS-CoV-2 and the development of neutralizing antibodies is the leading candidate for a correlate of protection. Non-

neutralizing antibody responses mediated by the crystallizable fragment (Fc) region also are critical in COVID-19 pathogenesis[4,5], with prolonged activation of the complement cascade contributing to tissue damage and symptoms of long COVID[6].

Epidemiological and vaccine studies have shown that anti-Spike (S) IgG, anti-S-receptor-binding domain (S-RBD) IgG, and neutralizing

antibodies correlate with protection against SARS-CoV-2[7,8]. The role of antibodies in the control of SARS-CoV-2 infection and the pathogenesis of the disease is still ambiguous as studies have consistently shown that both binding and neutralizing antibody titers are greater in patients with more severe COVID-19[9,10]. The greater magnitude of antibody titers is observed in severe COVID-19 patients both during the acute phase of the disease and convalescence[9,11]. The association of hospitalization and subsequent deaths in individuals with greater antibody responses raises questions about the role of antibodies in the protection versus pathogenesis of COVID-19. One study highlighted that the antibody repertoire in mild COVID-19 patients exhibits greater diversity, antibody class switching, and affinity maturation than in severe COVID-19 patients[12]. Despite having higher antibody titers, individuals with severe COVID-19 produce less potent and functional antibodies, thereby contributing to pathogenesis[13].

Despite known variations in the quantity and quality of antibody responses based on disease severity, the antibody dynamics that predict COVID-19 progression (i.e., survival or death) are still unclear. Most studies typically measure antibody responses in serum or plasma, but mucosal immunity to SARS-CoV-2, either in respiratory or oral fluid samples, may provide a better correlate of protection. Using a longitudinal cohort at Johns Hopkins Hospital, we analyzed antibody responses in plasma and mucosal samples, measured proinflammatory cytokines and chemokines in plasma, and determined the associations of key demographic variables and antibody responses at enrollment with COVID-19 outcomes. Using machine learning algorithms, we identified that serological variables, particularly anti-neucleocapsid (N) IgG titer, and anti-S complement C1q, are better predictors of intubation or death in COVID-19 patients, than socio-demographic variables.

## Methods

### Study cohorts
A convenience sample of hospitalized (n = 105) and non-hospitalized (n = 73) patients were enrolled in a prospective cohort study from April 2020 through April 2021 (Table 1). This study, including its recruitment and consenting process, was approved by the Institutional Review Board (IRB) of the Johns Hopkins University (IRB00245545, IRB00259948)[14–16]. Non-hospitalized individuals were screened and read an oral consent script over the phone for eligibility, during which research personnel ensured the participants understood the purpose and procedures of the study and recorded their verbal statement of consent. When enrolled participants were sent a study kit, a copy of the consent script was included. For hospitalized patients, written consent was obtained from all hospitalized participants. The study comprised Johns Hopkins Hospital in- or out-patients who were 18 years or older with reference lab RT-PCR-confirmed SARS-CoV-2 diagnosis. Blood plasma samples were collected from non-hospitalized patients at one-month post-enrollment (MPE). For non-hospitalized patients, days from enrollment were calculated by taking the date of the positive PCR test (considered as enrollment) relative to the date of each sample collection. For hospitalized patients, the days from enrollment were calculated by taking each sample collection date relative to the first sample collection date collected upon hospital admission for each patient. The enrollment timepoint was set as 0 days post-enrollment. Hospitalized samples collected 21–28 days after the initial enrollment sample collection were used for the 1 MPE timepoint. Samples from hospitalized patients at 1 MPE were collected on average 28 ± 11 days after PCR confirmation and 26 ± 3 days after hospital enrollment. The 1 MPE for non-hospitalized patients ranged between 18 to 91 days after PCR-confirmation, averaging 46 ± 15 days. Non-hospitalized samples used for 1 MPE analysis was restricted to samples collected within 31- and 61-days post-PCR confirmation based on the mean ± SD days post-PCR. Antibody levels were comparable among non-hospitalized patients within this time frame (Supplementary Fig. 1a-b). Blood plasma samples were collected from hospitalized patients at study enrollment, 1 MPE, and until subsequent death or up until 100 days post-enrollment (DPE) (Supplementary Fig. 1c, d). Oropharyngeal (OP) and nasopharyngeal (NP) swab samples

were collected at enrollment for all hospitalized patients. Non-hospitalized patients were assigned World Health Organization (WHO) COVID-19 severity scores of 1–2, and moderate, severe, and deceased hospitalized COVID-19 patients were assigned WHO scores of 3–4, 5–7, and 8, respectively (Supplementary Table 1). For hospitalized patients, the severity scores used were the maximum severity scores during their hospital stay. Samples were processed on the same day of collection and stored at −80 °C until the time of the biological assays.

### Virus RNA levels
SARS-CoV-2 RT-PCR testing was performed on OP or NP swab samples using Abbott m2000 platform and Abbott RealTime SARS-CoV-2 assay (09N77-095, Abbott Molecular, IL, USA) per the manufacturer's instructions[16,17]. SARS-CoV-2 viral RNA levels (copies/mL) were calculated from qPCR Ct values using the standard curve.

### SARS-CoV-2 variant inference
A likely variant of SARS-CoV-2 was inferred for each patient using the date of sample collection and the timeframe of variants during which community prevalence was above 95% according to Robinson, et al.[18]. The ancestral variant was prevalent from January 18, 2021, to July 31, 2021.

### Cytokine/chemokine detection
Plasma proinflammatory cytokines and chemokines were measured using a custom multiplex kit from Meso Scale Discovery (MSD; Rockville, MD) according to the manufacturer's instructions[15,19]. Cytokine and chemokine data were first shifted by a pseudo count of +1 to avoid zeros and then $\log_2$-transformed to have normal distributions. Analytes with signal below the background were set to 0 and lower limits of detection were based on the manufacturer's recommendations.

### Binding antibody measurement by ELISA on plasma samples
Binding antibodies in plasma samples were determined using in-house ELISAs[9,20,21]. The 96-well plates (Immulon 4HBK, Thermo Fisher Scientific) were coated overnight at 4 °C with 50 µL of 2 µg/mL of either Spike (S), spike receptor binding domain (S-RBD), or 1 µg/mL of nucleocapsid (N) antigen diluted in 1X phosphate-buffered saline (PBS). Antigens were either engineered at Johns Hopkins University[9] or were obtained through the National Cancer Institute Serological Sciences Networks (SeroNet) for COVID-19[22]. Plates were washed 3 times with 200 µL of wash buffer (PBS with 0.1% Tween-20) and then blocked with 3% milk powder in PBS with 0.1%Tween-20 (PBS-T) for 1 h at room temperature (RT). Heat-inactivated plasma samples were three-fold serially diluted 10 times, starting with 1:20 dilution in dilution buffer (1% milk + 0.1% PBS-T). The blocking buffer was removed and 100 µL of diluted plasma samples were transferred. Plates were incubated for 2 h at RT, washed, and 50 µL of anti-human HRP IgG (1:5000, #A18823, Invitrogen, Thermo Fisher Scientific), IgA (1:5000, #A18787, Invitrogen, Thermo Fisher Scientific), IgG1 (1:4000, #9054-05, Southern Biotech), IgG2 (1:4000, #9060-05, Southern Biotech), IgG3 (1:4000, #9210-05, Southern Biotech) or IgG4 (1:8000, #9200-05, Southern Biotech) secondary antibody was added. After 1 h incubation at RT, plates were washed, and 100 µL of Sigmafast OPD (o-phenylenediamine dihydrochloride) solution (MilliporeSigma) was added. After 10 minutes of incubation at RT, the reaction was stopped by adding 50 µL of 3 M HCL (Thermo Fisher Scientific) and the plates were read for OD values at 490 nm wavelength on a SpectraMax i3 ELISA plate reader (BioTek Instruments). Background-subtracted optical density values were plotted against the dilution factor to calculate the area under the curve (AUC). Spike and N IgG antibodies were converted into the international binding assay units (BAU/mL) using the standards calibrated at the Johns Hopkins University through the SeroNet assay harmonization project[22]. AUC and BAU/mL values were log-transformed for analysis. The limit of detection (LOD) was determined as half of the lowest BAU for the sample with a detectable titer (i.e., titer ≥20), while samples with undetectable titers (i.e., <20) received a value that was half the limit of detection[20].

**Table 1 | COVID-19 patient demographics and comorbidities at enrollment**

| COVID-19 severity category | | | | | |
|---|---|---|---|---|---|
| | **Non-Hospitalized (1–2)** | **Hospitalized-Moderate (3–4)** | **Hospitalized-Severe (5–7)** | **Hospitalized-Deceased (8)** | **Total** |
| Sex (n, %) | | | | | |
| Female | 46 (63%) | 18 (44%) | 20 (50%) | 10 (41.7%) | 94 (53%) |
| Male | 27 (37%) | 23 (56%) | 20 (50%) | 14 (58.3%) | 84 (47%) |
| Race/Ethnicity (n, %) | | | | | |
| White | 36 (49%) | 12 (29%) | 14 (35%) | 4 (17%) | 66 (37%) |
| Black | 23 (32%) | 21 (51%) | 20 (50%) | 12 (50%) | 76 (43%) |
| Asian | 1 (1%) | 1 (2%) | 1 (2.5%) | 1 (4%) | 4 (2%) |
| Other | 13 (18%) | 6 (15%) | 5 (12.5%) | 7 (29%) | 31 (17%) |
| N/A | 0 (0%) | 1 (2%) | 0 (0%) | 0 (0%) | 1 (1%) |
| Age, mean (SD) | 51 (15) | 54.7 (13.6) | 58.15 (12.1) | 62.3 (12.3) | 55 (13.9) |
| BMI, mean (SD) | 31.4 (8.8) | 31.5 (6.9) | 32.3 (8.9) | 33.5 (10.1) | 32 (8.6) |
| Intubated (n, %) | 0 (0%) | 0 (0%) | 24 (60%) | 24 (100%) | 48 (27%) |
| HIV (n, %) | 2 (3%) | 3 (7%) | 1 (3%) | 1 (4%) | 7 (4%) |
| Pulmonary disease (n, %) | 22 (30%) | 10 (24%) | 12 (30%) | 6 (25%) | 50 (28%) |
| Diabetes (n, %) | 11 (15%) | 17 (41%) | 16 (40%) | 10 (42%) | 54 (30%) |
| Autoimmune disease (n, %) | 8 (11%) | 5 (12%) | 4 (10%) | 1 (4%) | 18 (10%) |
| Cancer (n, %) | 8 (11%) | 14 (34%) | 9 (23%) | 7 (29%) | 38 (21%) |
| Organ Transplant (n, %) | 3 (4%) | 3 (7%) | 4 (10%) | 1 (4%) | 11 (6%) |
| Total | 73 (41%) | 41 (23%) | 40 (22%) | 24 (13%) | 178 (100%) |

## ACE2 binding inhibition antibody assay

ACE2 binding inhibition antibody assay was performed using MSD V-PLEX SARS-CoV-2 ACE2 kits (Panel 29) according to the manufacturer's protocol[20]. Antigen pre-coated plates were washed and incubated with plasma samples (1:100 dilution) for 1 h followed by the addition of SULFO-TAG conjugated human ACE2 protein for 1 h at RT. After incubation, plates were washed, buffer was added, and plates were read with a MESO QuickPlex SQ 120 instrument. ACE2 binding inhibition activity corresponding to 1 µg/mL of monoclonal antibody to the ancestral strain of SARS-CoV-2 S protein was determined using an 8-point calibration curve included in each plate. Percent inhibition was determined based on the equation ([1 – average sample electrochemiluminescence/average electrochemiluminescence signal of blank well] × 100) provided by the manufacturer.

## Complement activation assay

Complement activation assays were performed from plasma samples as described[23] with modifications. Nunc MaxiSorp flat-bottom 96-well plates were coated with 100 ng/well of S, S-RBD, or PBS alone. After overnight incubation, plates were washed with 0.1% PBS-T and blocked with 1% gelatin/PBS-T for 1 h at RT. 100 µL of heat-inactivated patient plasma diluted at 1:1000 in 1% gelatin/PBS-T were added to the wells and incubated for 1 h at RT. After washing with PBS-T, normal human serum (NHS, Comptech) at 1:50 dilution in gelatin veronal buffer with calcium and magnesium (GVB++, Comptech) was added as the complement source. To remove any background anti-spike IgG response, total IgG was removed from NHS source. NHS was diluted 1:50 in GVB++ and incubated with increasing amounts of PureProteome protein A/G mix magnetic beads (Millipore) for 1 h at 4 °C with continuous mixing. Total IgG and anti-S background antibodies were fully removed, without affecting complement activity, using 50 µL beads per 300 µL of diluted NHS. After 1 h incubation with NHS at 37 °C, wells were washed with PBS-T, and goat anti-human C1q (Comptech, A200) diluted 1:20,000 in PBS-T was added for 1 h at RT. HRP-labeled anti-goat IgG (Thermo Fisher Scientific, A16005) diluted 1:5000 in PBS-T was used as secondary antibody and incubated for 1 h at RT. Following addition of SureBlue peroxidase reagent (IPL), reactions were stopped with HCL and absorbances were read at 450 nm. Arbitrary units (AU) were calculated using a standard minus background binding to PBS-coated wells.

## Antibody-dependent cell-mediated cytotoxicity (ADCC) assays

The spike-expressing cell lines were generated by transfecting Tet-on HEK293 (ATCC, CRL-1573) cell lines with Sleeping Beauty-based transposons designed to express (a) PuroR antibiotic resistance gene and (b) one or another form of the SARS-CoV-2 spike protein[24]. Cell line identity was confirmed by anti-spike immunoblot of cells grown in the absence or presence of doxycycline, and by diagnostic PCR but no mycoplasma testing was performed. Tet-on HEK-293 cells engineered to express Wuhan-1 S protein (hereafter HtetZ/SW1 HEK293 cells)[24] in response to doxycycline (DOX) were incubated overnight with 1 µg/mL DOX in DMEM containing 10% fetal bovine serum (FBS), 1% penicillin/streptomycin (P/S), zeocin (200 µg/mL) and puromycin (3 µg/mL). HtetZ/SW1 HEK-293 cells were detached with trypsin/EDTA, resuspended at $2 \times 10^6$ cells/mL in Iscove's Modified Dulbecco's Medium (IMDM) (10% FBS and 1% P/S). ADCC assays were performed in 96-well round bottom plates by incubating 50 µL HtetZ/SW1 HEK-293 cells with 1 µg of IgG purified from patient plasma (Melon Gel Spin Plate Kit, Thermo Fisher Scientific). After 30-minutes at 37 °C, 50 µL of Jurkat-Lucia™ NFAT-CD16 cells (InvivoGen) at $4 \times 10^6$ cells/mL were added per well (effector: target ratio 2:1), mixed and centrifuged for 1-minute, at 800 rpm. After 5-h incubation at 37 °C, 20 µL of supernatant was collected and mixed with 50 µL of QuantiLuc™ solution (InvivoGen) in a 96-well black polystyrene plate (Corning Costar) to assess luciferase activity. A pool of high titer anti-S IgG purified from patient plasma was used to generate a standard curve to calculate the unknown sample AU and calibrate across plates.

## Flow cytometry

Spike surface expression for ADCC assays was confirmed by flow cytometry using commercial SARS-CoV-2 2019-nCoV spike S2 antibody (Sino Biological 40590-D001) and using purified IgG from anti-S positive patient plasma (n = 3; Supplementary Fig. 2). 50 µL of 500,000 HEK293 cells (HtetZ/CG145 (SpikeW1) HEK293 or HtetZ HEK293) were resuspended in

staining solution (PBS with 2% fetal bovine serum) and incubated with either Sino Biological 40590-D001 (1:500 dilution) or anti-Spike Positive Patient Plasma (aS PPP) purified IgG (10 μg/mL and 40 μg/mL concentrations) for 30 min on ice in the dark. The cells were then washed twice with staining solution and incubated for 30 min with the respective secondary antibodies (BV421 anti-mouse IgG Ab, Biolegend 405317; AF488 anti-human IgG Ab, ThermoFisher A11013, all at 2 μg/mL concentration). Afterwards, the cells were washed twice and resuspended in 500 μL of staining solution. The counts of BV421-labeled SB40590 or AF488-labeled aS PPP IgG cells were measured using the BD FACSAria II Cell Sorter (Supplementary Table 3) and analyzed with Kaluza Analysis software version 2.2.1 and Inkscape.

## Multiplex antibody assays on mucosal samples

IgG and secretory IgA (sIgA) antibody responses on NP and OP swabs were determined using multiplex SARS-CoV-2 antibody assays[25–27]. The SARS-CoV-2 multiplex assay included two SARS-CoV-2 N antigens, two S, three S-RBD antigens, endemic coronavirus OC43, NL63, HKU1 and 229E antigens, respiratory syncytial virus (RSV), and several control beads (total IgG, IgA, IgM, BSA). Mucosal samples were added to the assay buffer (PBS with 0.05% Tween 20 and 0.1% BSA) containing 1000 beads per bead set in each well of a 96-well plate. NP and OP swabs were tested at a 1:2 dilution for IgG and a 1:4 for sIgA. After a 1 h sample incubation beads were washed twice, then phycoerythrin (PE)-labeled anti-human IgG or mouse anti-secretory component antibody, followed by PE-labeled anti-mouse antibody was added. After another 1-h incubation beads were washed twice again and then read on a MagPix.

## Statistics and reproducibility

All antibodies (i.e., mucosal and serum antibodies measured as either AUC, BAU/mL, AU, or MFI) and virus RNA (copies/mL) data were $\log_{10}$-transformed. To account for possible zeros, complement and ADCC data (AU) were shifted by +1 prior to logarithmic transformation. ACE2 inhibition data (%) were arcsine transformed to be more consistent with the Gaussian assumptions used in analyses. Non-hospitalized and hospitalized serological data at 1 MPE were compared using linear regression analysis, controlling for biological sex and age. The null hypotheses that hospitalization group means were equal at each time point were tested using Welch's ANOVA with Benjamini-Hochberg post-hoc corrections at a 0.1 false discovery rate (FDR). Spearman correlation was used to quantify the association of viral load between nasal and oral samples and the association of complement C1q with binding antibodies. Linear mixed-effects regression modeling was used to compare antibody trajectories over days from enrollment across COVID-19 disease severity groups among hospitalized. The results were visualized by plotting the estimated fixed effects against days since enrollment for different severity groups. The null hypothesis that all groups had the same dependence on time was tested using a likelihood ratio test comparing mixed effects models with and without the group by time interaction. Binding, complement, and ACE2 inhibition antibody data were scored by quartiles from 0 to 3 with data in the lower 25th percentile scored as 0 and those in upper 75th percentile scored as 3. Data were then totaled by antibody type (e.g., anti-N IgG, anti-S IgG, anti-S-RBD IgG, and anti-S-RBD IgA quartile scores were totaled by participant for an overall binding antibody score) to create an index score. Logistic regression models, with death as the binary outcome, were used against antibody scoring to evaluate how antibody levels were associated with the probability of death at enrollment or 1 MPE. From the enrollment data, complete datasets without missing data were available for 98 hospitalized patients for 24 variables. We implemented two approaches for random forest modeling—one using out-of-bag (OOB) predictions without data partitioning and another using stratified 10-fold cross-validation with data partitioning of 90% for training and 10% for testing. Random forest algorithms were used to compare the predictive power of sociodemographic (i.e., age, BMI, race/ethnicity, sex, and clinical comorbidities) and serological variables for intubation or death as represented by the variable importance plots. The performance of

random forest models was assessed using the area under the receiver operating characteristic curves (AUROC) for out-of-bag predictions (OOB)[28] or 10-fold cross-validation predictions. Sensitivity (true positive), specificity (true negative), error rates, and F1 scores were calculated based on the confusion matrix at a probabilistic cutoff of 0.46 and 0.21 for the intubation and death models, respectively. Random forest models using only demographic, serological, or both types of measures were compared by AUROC metrics. AUROC values closer to 1 indicate a higher quality of model performance with a more accurate classification of the data. All random forest models were trained using the same seed value to ensure reproducibility. randomForest, caret, ROCR, MLmetrics, and pdp packages in R were used for the random forest modeling, calculating variable importance scores, evaluating performance metrics, and visualizing dependence plots[29–33]. All p-values < 0.05 were considered statistically significant. Statistical analyses were conducted in Stata 18.0, GraphPad Prism 10, and RStudio 2022.7.2.576[34,35].

## Reporting summary

Further information on research design is available in the Nature Portfolio Reporting Summary linked to this article.

## Results

### Demographic characteristics of the COVID-19 study cohorts

A total of 73 (46 female; 27 male) non-hospitalized and 105 (48 female; 57 male) hospitalized COVID-19 patients were included (Table 1). Out of the hospitalized patients, 41 (18 female, 23 male) were in the WHO moderate disease category, 40 patients (20 female; 20 male) were in the severe disease category, and 24 patients (10 female; 14 male) were deceased. For non-hospitalized patients, samples were collected for the '1 MPE' timepoint at 46 ± 15 days and neither anti-N IgG nor anti-S IgG responses correlated with the number of days post-enrollment (Supplementary Fig. 1a, b). Sample collection from hospitalized patients at the 1 MPE timepoint averaged at 28 ± 11 days after PCR confirmation and 26 ± 3 days after hospital enrollment (Supplementary Fig. 1c, d). The number of days from hospital enrollment to death among the deceased cohort ranged between 3 to 261 days, with 75% of those dying from COVID-19 within 66 days of enrollment (Supplementary Fig. 1e). For longitudinal analyses and predictive modeling, data from hospitalized patients who died within 100 DPE (n = 18) were included.

### Proinflammatory cytokine/chemokine, but not viral RNA, levels at enrollment are greater among hospitalized patients with more severe COVID-19

Virus RNA quantification was performed in OP and NP swabs collected from the hospitalized patients during enrollment. Viral RNA copy numbers did not differ among moderate, severe, and deceased patients in either NP or OP swab samples (p > 0.05, Fig. 1a, b. Virus RNA levels in OP and NP swabs were positively correlated (Spearman R = 0.659, p = 0.0012, Fig. 1c). During enrollment, inflammatory cytokine/chemokine response levels in plasma were compared among hospitalized patients with different COVID-19 disease severities (Supplementary Table 2). Consistent with previous reports[36,37], patients with severe disease (WHO score 5–7) or those who subsequently died from COVID-19 (WHO score 8) had greater concentrations of proinflammatory cytokines and chemokines, including IL-6, IL-8, TNF-α, IL-15, IL-16, and MCP-1, than hospitalized patients with moderate disease (WHO score 3–4) (p < 0.05 in each case, Fig. 1d–i).

### COVID-19 disease severity is not associated with mucosal antibody responses in hospitalized patients

Using the OP and NP swab sample viral transport media collected during hospital enrollment, ancestral SARS-CoV-2 N- and S-specific IgG and secretory IgA (sIgA) antibody responses were measured. Binding antibody responses in mucosal samples against ancestral viral antigens did not differ based on COVID-19 disease severity among hospitalized patients (p > 0.05 in each case, Fig. 2a–h). The sIgA (Supplementary Fig. 3a–f) and IgG

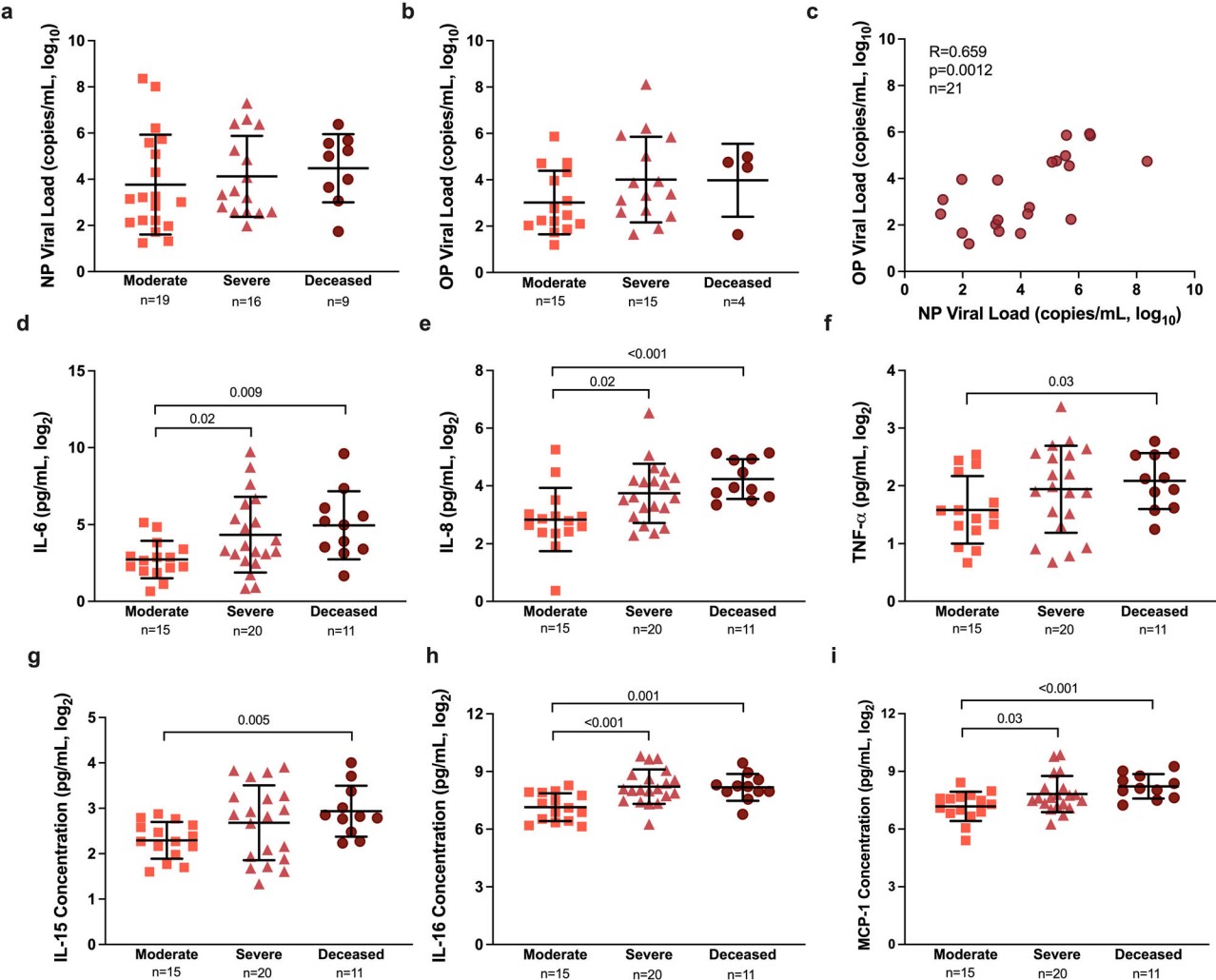

**Fig. 1 | SARS-CoV-2 virus RNA and cytokine/chemokine responses among hospitalized COVID-19 patients at enrollment. a** Nasopharyngeal (NP) viral load (copies/mL, $\log_{10}$) and (**b**) oropharyngeal (OP) viral load (copies/mL, $\log_{10}$) were measured by qPCR at enrollment and compared among patients classified as moderate (WHO score 3–4), severe (WHO score 5–7), or deceased (WHO score 8). **c** The Spearman correlation between OP and NP viral loads at enrollment.

**d–i** Concentrations (pg/ml) of several proinflammatory cytokines and chemokines that differed among COVID-19 hospitalized patients classified as moderate, severe, or deceased. Data are presented as means with standard deviations, indicated by error bars. *p*-values for statistically significant differences (p < 0.05) by Welch's ANOVA are shown in the figures.

(Supplementary Fig. 4a–f) responses also were measured against other beta coronaviruses, including SARS, MERS, HCoV, and OC43, in the NP and OP swab samples and were not significantly different among moderate, severe, and deceased patients in both OP and NP compartments. These data suggest mucosal antibody responses against SARS-CoV-2 do not differ by COVID-19 severity.

**Antibody responses are higher among hospitalized than non-hospitalized COVID-19 patients at 1 MPE**

Using plasma samples collected at 1 MPE, we compared antibody binding (i.e., anti-S IgG, anti-S-RBD IgG, anti-S-RBD IgA, and anti-N IgG), ACE2 binding inhibition, and Fc effector antibody responses (i.e., complement activation and ADCC) between non-hospitalized and hospitalized patients. Binding antibodies (Fig. 3a–d) were significantly higher (p < 0.05) among hospitalized patients than non-hospitalized patients at 1 MPE. Likewise, ACE2 binding inhibition antibody responses were significantly higher among hospitalized than non-hospitalized patients (p = 0.000, Fig. 3e). The Fc effector antibody functions, including complement activation as measured by C1q binding to surface-bound anti-S and anti-S-RBD antibodies (hereafter anti-S C1q and anti-S-RBD C1q,

respectively), and ADCC, were significantly higher (p < 0.05 in each case) in hospitalized than non-hospitalized patients (Fig. 3f–h). Neither the reported sex (Supplementary Fig. 5) nor age (Supplementary Fig. 6) of the patients impacted binding, ACE2 binding inhibition, or Fc effector antibody responses among either non-hospitalized or hospitalized patients at 1 MPE in this cohort. Consistent with previous findings[9,10], people who required hospitalization for acute COVID-19 had higher antibody responses at 1 MPE than patients who did not require hospitalization (Fig. 3, Supplementary Figs. 5–6).

**COVID-19 disease severity is correlated with plasma antibody responses over time until death or 100 DPE**

Binding and ACE2 binding inhibition antibodies were measured in plasma samples from hospitalized patients, collected at hospital enrollment and through subsequent death or 100 DPE. During enrollment, anti-S IgG, but not anti-S-RBD IgG, anti-S-RBD IgA, anti-N IgG, nor ACE2 binding inhibition antibody responses, were significantly higher among patients with severe compared to moderate disease (p = 0.007, Fig. 4a–e). After 1 MPE, anti-S IgG (Fig. 4a), anti-S-RBD IgG (Fig. 4b), and ACE2 binding inhibition (Fig. 4e) antibody responses significantly increased over time

**Article**

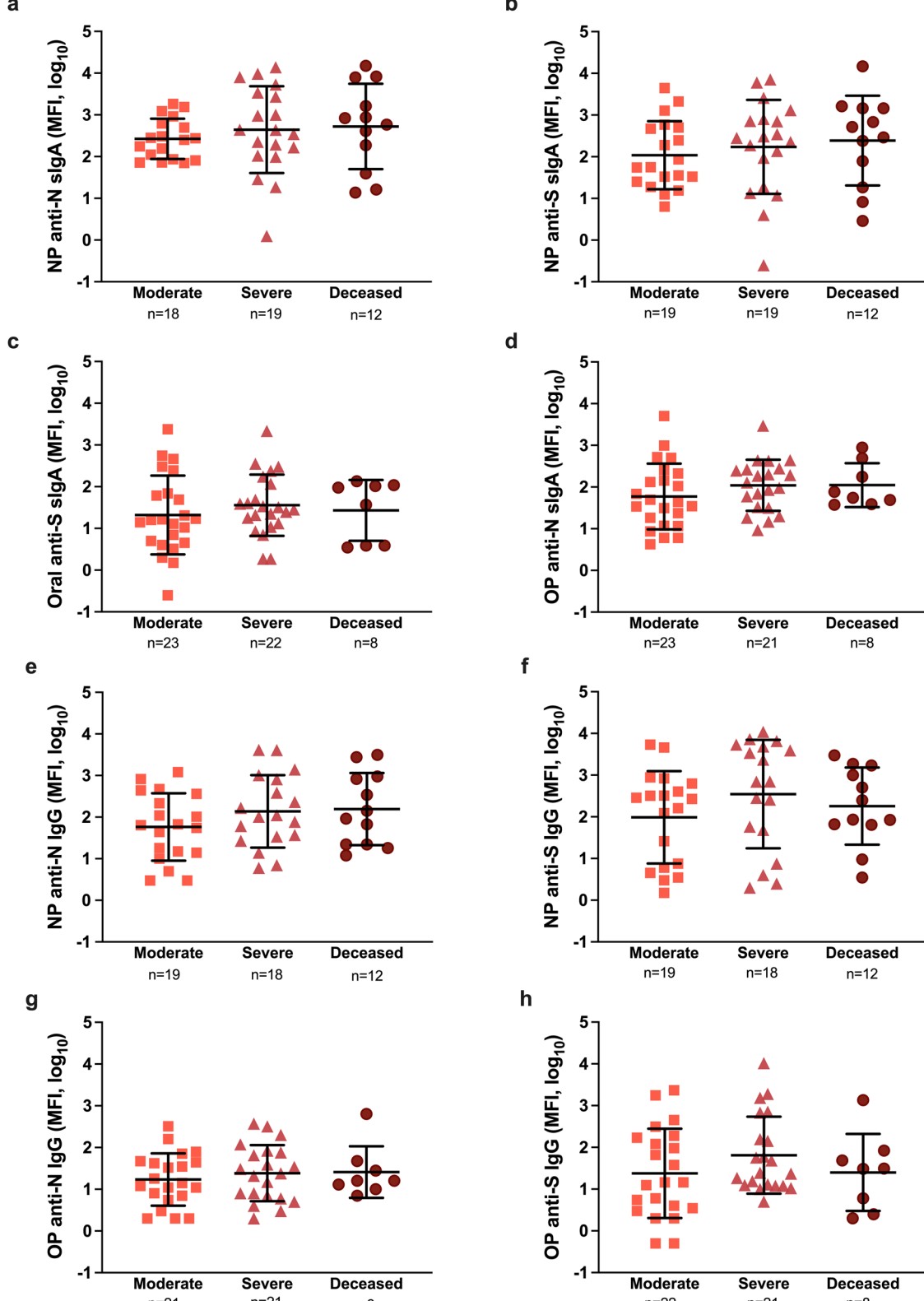

**Fig. 2 | Mucosal antibody responses among hospitalized COVID-19 patients at enrollment. a–d** Anti-nucleocapsid (N) and anti-spike (S) secretory IgA or (**e–h**) IgG responses were measured as median fluorescence intensity (MFI) in naso-pharyngeal (NP) or oropharyngeal (OP) samples and compared among COVID-19 hospitalized patients classified as moderate (WHO score 3–4), severe (WHO score 5–7), or deceased (WHO score 8). Data are analyzed using Welch's ANOVA and presented as means with standard deviations, indicated by error bars.

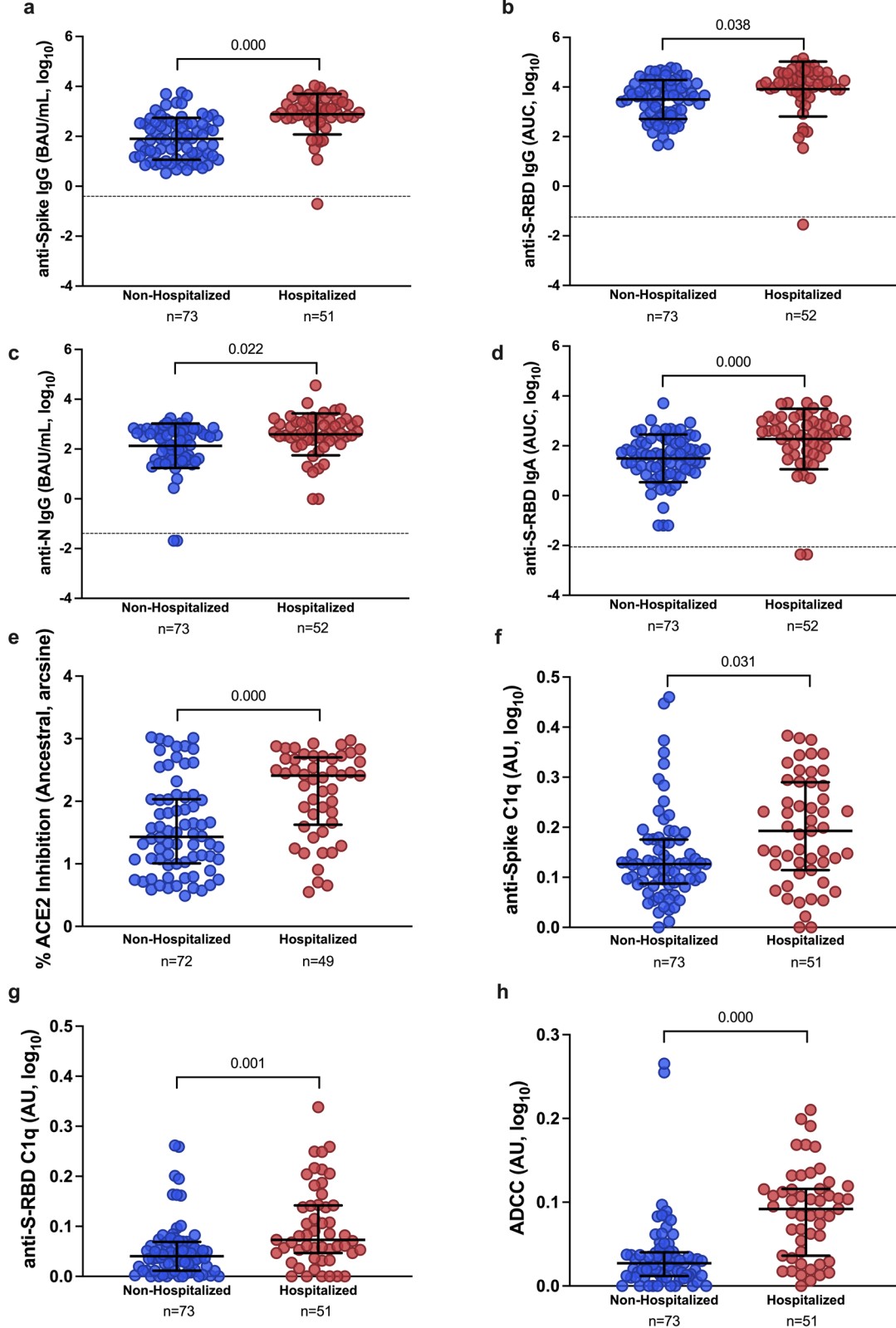

**Fig. 3 | Antibody responses in plasma samples of non-hospitalized and hospitalized COVID-19 patients at 1-month post-enrollment (MPE). a–d** IgG binding antibody responses against ancestral spike (S), spike receptor binding domain (S-RBD), and nucleocapsid (N) were quantified by ELISA and calculated as the binding antibody units (BAU) per ml if international standards were available or as the area under the curve (AUC) if standards were not available and titration curves only could be generated; (**e**) ACE2 binding inhibition antibodies were measured using MSD V-PLEX SARS-CoV-2 ACE2 kits; and (**f–h**) Fc effector antibody responses were quantified using complement fixation and antibody-dependent cellular cyto-toxicity (ADCC) assays. All assays were run using ancestral SARS-CoV-2. Data were compared using linear regression analysis, controlling for age and biological sex, to look at differences between unvaccinated non-hospitalized and hospitalized patients at 1 MPE. Data are presented as means with standard deviations, indicated by error bars. The limit of detection (LOD) is indicated by the dashed lines. *p*-values for statistically significant differences (p < 0.05) are shown in the figures.

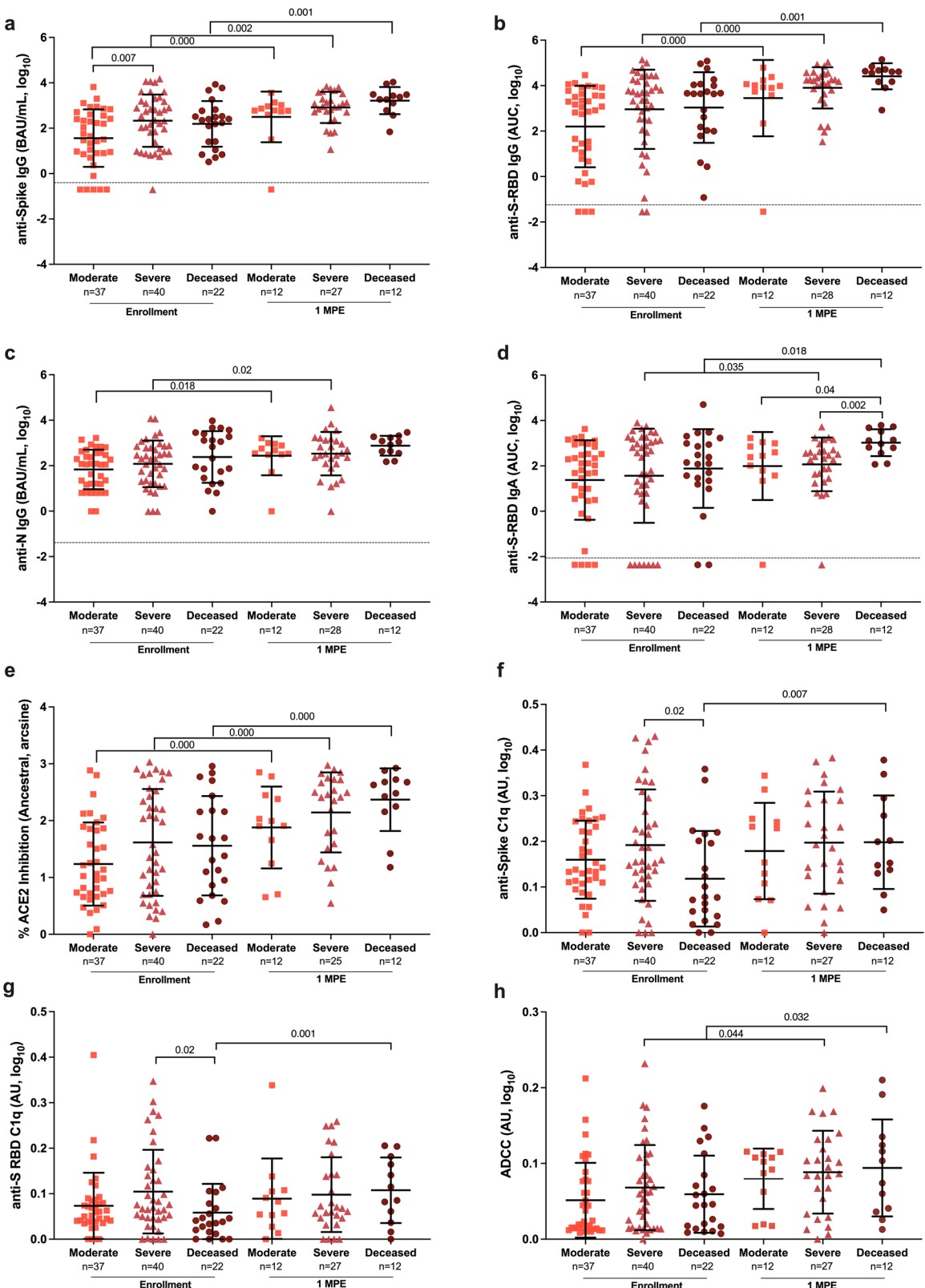

(p < 0.05 in each case) among all hospitalized patients, with the deceased patients consistently maintaining the highest antibody responses. Anti-N IgG responses increased at 1 MPE among both severe (p = 0.018) and moderate (p = 0.02) disease patients but did not change among deceased patients (p = 0.06, Fig. 4c). Among hospitalized patients with severe disease or dying from COVID-19, anti-S-RBD IgA (Fig. 4d) increased over time

since enrollment. At 1 MPE, patients who died from COVID-19 had significantly greater anti-S-RBD IgA response than patients with moderate or severe disease (p < 0.05 in each case). Unlike ancestral SARS-CoV-2 (Fig. 4e), ACE2 inhibition antibodies against SARS-CoV-2 variants were comparable among hospitalized patients with varying severities of disease (Supplementary Fig. 7).

**Fig. 4 | Binding, ACE2 inhibition, and Fc effector antibody responses in plasma among COVID-19 hospitalized patients at enrollment and 1-month post-enrollment (MPE).** The binding (**a–c**) IgG and (**d**) IgA antibodies recognizing ancestral SARS-CoV-2 spike (S), spike receptor binding domain (S-RBD), or nucleocapsid (N) were quantified by ELISA, and measured as the binding antibody units (BAU) per mL if international standards were available or as the area under the curve (AUC) if standards were not available and titration curves could only be generated. **e** The percentage of ACE2 inhibition for the ancestral SARS-CoV-2 variant was calculated and arcsine transformed for analyses. **f–h** The Fc effector antibody responses were measured based on C1q complement fixation in response to either the spike or S-RBD or antibody dependent cellular cytotoxicity and reported as arbitrary units (AU). Antibody responses were compared among COVID-19 hospitalized patients classified as moderate (WHO score 3–4), severe (WHO score 5–7), or deceased (WHO score 8) using samples collected at enrollment vs. 1 MPE. Data are presented as means with standard deviations, indicated by error bars. $p$-values for statistically significant differences ($p < 0.05$) by linear mixed-effects regression to compare change over time or Welch's ANOVA to compare across groups within a time point are indicated. Limit of detection (LOD) are indicated by the dashed lines.

Because subclasses of IgG have different antibody effector functions, subclasses of IgG recognizing SARS-CoV-2 S were analyzed. At enrollment, anti-S IgG2 and IgG3 were significantly higher among either deceased or severe disease patients than moderate disease patients (Fig. 5). From enrollment to 1 MPE, anti-S IgG1 and IgG3 levels significantly increased among all hospitalized patients, whereas anti-S IgG2 and IgG4 only increased over time among patients with moderate disease or those who died from COVID-19.

Because differential Fc effector antibody functions that mediate complement and innate immune cell activation can contribute to COVID-19 pathology[38–40] and predict symptoms of long COVID[6], anti-S C1q, anti-S-RBD C1q, and ADCC were measured using antibodies from plasma samples collected from hospitalized patients at enrollment and 1 MPE (Fig. 4f–h). At enrollment, anti-S C1q and anti-S-RBD C1q (Fig. 4f, g) were significantly lower ($p < 0.05$) in the patients who died from COVID-19 compared to hospitalized patients with severe disease. In contrast, ADCC responses were not significantly different among moderate, severe, and deceased patients (Fig. 4h). Only deceased COVID-19 patients had a significant increase in anti-S C1q ($p = 0.007$, Fig. 4f) and anti-S-RBD C1q activities ($p = 0.001$, Fig. 4g) over time, from enrollment to 1 MPE. There was a significant increase in ADCC responses from enrollment to 1 MPE in patients with either severe disease ($p = 0.044$) or who died from COVID-19 ($p = 0.032$, Fig. 4h). The complement activity was primarily mediated by IgG rather than IgM antibodies as shown by the stronger correlation of complement with IgG than IgM (Supplementary Fig. 8a–h). IgM antibodies, however, were better correlated with complement activity among hospitalized than non-hospitalized patients. Anti-S IgG1 and IgG3, but not anti-S IgG2 or IgG4, strongly correlated with anti-S C1q and anti-S-RBD C1q among hospitalized patients (Supplementary Fig. 8i–p).

With consideration of the antibody kinetics from days since enrollment until either death or 100 DPE among hospitalized COVID-19 patients, anti-S IgG, anti-S-RBD IgG, anti-N IgG, anti-S-RBD IgA, and ACE2 binding inhibition (Fig. 6a–e) were maintained at higher levels over time among deceased patients as compared to other hospitalized patients. Fc effector activities, including complement activation and ADCC, exhibited no changes over time among hospitalized patients (Fig. 6f–h).

## Predictive value of plasma antibody titer as a biomarker for COVID-19-related death among hospitalized patients

We sought to understand the predictive value of antibody titers as a biomarker for subsequent death from COVID-19 among hospitalized patients (Fig. 7a–h). A cumulative antibody score was calculated by first dividing each antibody measure into quartiles with assigned scores of 0 to 3, ranging from the lowest quartile to the highest quartile, and totaled across the measures by type of response (e.g., binding antibody index score is the sum of the quartile scores across anti-N IgG, anti-S IgG, anti-S-RBD IgG, and anti-S-RBD IgA). Using logistic regression modeling with death as a binary outcome against antibody scoring, greater cumulative binding antibody scores at 1 MPE were associated with an increased probability of death due to COVID-19 ($p = 0.045$; Fig. 7e), which was not observed at enrollment ($p = 0.39$; Fig. 7a). Similarly, a positive, but not statistically significant, association between the probability of death and ACE2 binding inhibition antibody scoring was observed at 1 MPE ($p = 0.17$; Fig. 7f), but not at enrollment ($p = 0.508$; Fig. 7b). The ability of anti-S antibodies to induce ADCC at either enrollment ($p = 0.87$; Fig. 7c) or at 1 MPE ($p = 0.94$; Fig. 7g) was not associated with death from COVID-19. Antibody-induced complement activation during enrollment ($p = 0.014$; Fig. 7d), but not at 1 MPE ($p = 0.75$; Fig. 7h), was negatively associated with the probability of death due to COVID-19. Logistic regression models cannot establish a causative relationship between binding antibody levels or complement with subsequent death outcomes among hospitalized patients, but rather demonstrate an association that should be further investigated.

Random forest models were used to evaluate sociodemographic (e.g., age, sex, BMI, race/ethnicity), clinical comorbidities, and serological measures at enrollment as predictors of subsequent intubation or death among hospitalized patients. Using complete data from 98 hospitalized patients ($n = 45$ intubated and $n = 21$ with subsequent death), the intubation model, comparing hospitalized patients who were intubated or not, had an AUROC value of 0.74 (Supplementary Fig. 9) and, similarly, the model for death had an AUROC value of 0.70 (Supplementary Fig. 10). For both intubation and death models, anti-N IgG antibodies and anti-S antibody-mediated complement fixation (anti-S C1q) were consistently prioritized as top variables that predicted intubation or death with the greatest mean decrease accuracy according to variance importance plots (Fig. 7i, j). Partial and bivariate dependence plots revealed that higher anti-N IgG titers and lower anti-S-mediated complement activation at enrollment were associated with greater predicted probabilities of intubation or death when controlling for all other variables (Supplementary Figs. 9–10). For the intubation model, anti-N IgG titers ranked first, anti-S IgG4 titers ranked second, anti-S C1q ranked third, and BMI ranked fourth for predictive ability and were the top variables necessary for accurately classifying patients as intubated based on enrollment data in our model (Fig. 7i). For death from COVID-19, anti-S C1q ranked first, anti-N IgG titer ranked second, anti-S-RBD C1q ranked third, and anti-S IgG titer ranked fourth for predictive ability (Fig. 7j). To further confirm these findings, we ran random forest models with either only sociodemographic variables or serological variables. The AUROC value for the random forest intubation model with only sociodemographic variables (0.55) was much lower than that of the random forest intubation model with only serological variables (0.68), indicating that performance of random forest models with only sociodemographic variables is inferior to those with serological measures in our cohort. Similarly, the stratified 10-fold cross-validation random forest models, including serological and sociodemographic variables, for intubation or death prioritized anti-N IgG and anti-S C1q as the top variables necessary for hospital outcome classification with AUROC values of 0.78 and 0.75, respectively (Supplementary Fig. 11). The models including serological variables consistently performed better compared to the models with only sociodemographic variables (Supplementary Fig. 11). Overall, our models suggest that serological variables, particularly anti-N IgG titer, and anti-S C1q, were better able to classify patients with subsequent intubation or death compared to sociodemographic variables.

## Discussion

In the current study, patients who became severely ill or died from COVID-19 consistently maintained greater antibody responses compared to hospitalized patients with moderate disease or non-hospitalized patients. We utilized samples collected from peripheral blood and mucosal sites to analyze over 20 different antibody characteristics, including diverse antibody

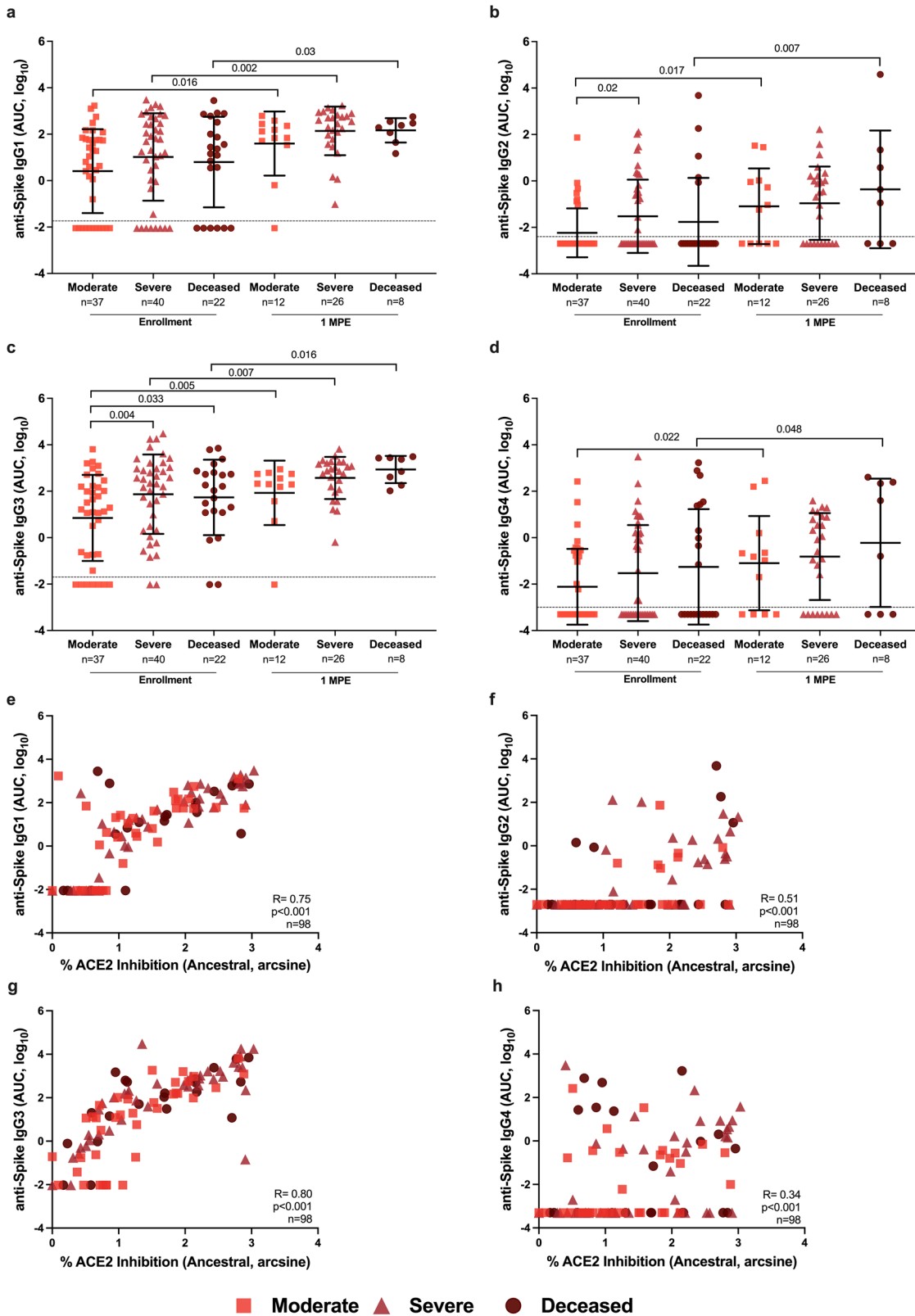

**Fig. 5 | Analysis of anti-Spike (S) IgG subclasses (IgG1-4) among hospitalized COVID-19 patients at enrollment and 1-month post-enrollment (MPE).** The binding of IgG1 (**a**), IgG2 (**b**), IgG3 (**c**), and IgG4 (**d**) to ancestral SARS-CoV-2 S antigen was measured as the area under the curve (AUC). Spearman correlation of IgG1 (**e**), IgG2 (**f**), IgG3 (**g**), and IgG4 (**h**) with % ACE2 inhibition at enrollment. Hospitalized COVID-19 patients were classified as moderate (WHO score 3–4), severe (WHO score 5–7), or deceased (WHO score 8). Data are presented as means with standard deviations, indicated by error bars. The limit of detection (LOD) is indicated by the dashed lines. $p$-values for statistically significant differences ($p < 0.05$) by linear mixed-effects regression to compare change over time or Welch's ANOVA to compare across groups within a time point are indicated.

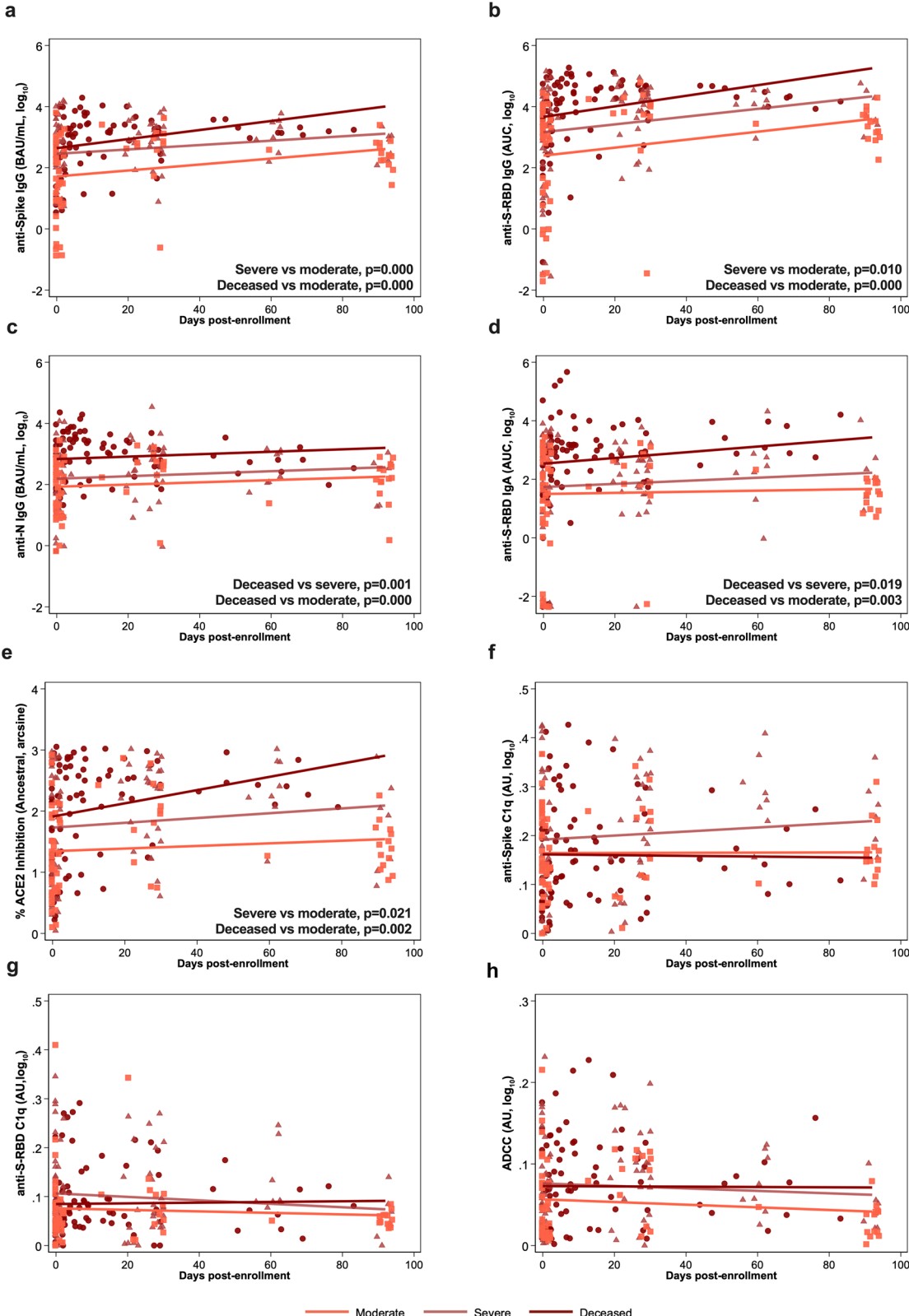

**Fig. 6 | Antibody responses against ancestral SARS-CoV-2 over continuous days since enrollment until 100 days post-enrollment (DPE) or subsequent death among hospitalized COVID-19 patients.** Linear mixed-effects regression models for (**a–d**) anti-spike (S), anti-spike receptor binding domain (S-RBD), or anti-nucleocapsid (N) IgG or IgA, measured as the binding antibody units (BAU) per ml if international standards were available or as the area under the curve (AUC) if standards were not available and only titration curves could be generated; (**e**) the percentage ACE2 inhibition against ancestral SARS-CoV-2 as a surrogate of virus neutralization, and (**f–h**) Fc effector antibody responses as measured by complement fixation against spike or S-RBD or antibody-dependent cellular cytotoxicity (ADCC) up until 100 DPE or death among hospitalized patients classified as moderate (WHO score 3–4; n = 41), severe (WHO score 5–7; n = 40), or deceased (WHO score 8; n = 24). *P*-values for statistically significant differences (p < 0.05) by linear mixed-effects regression contrasts are shown within the figures.

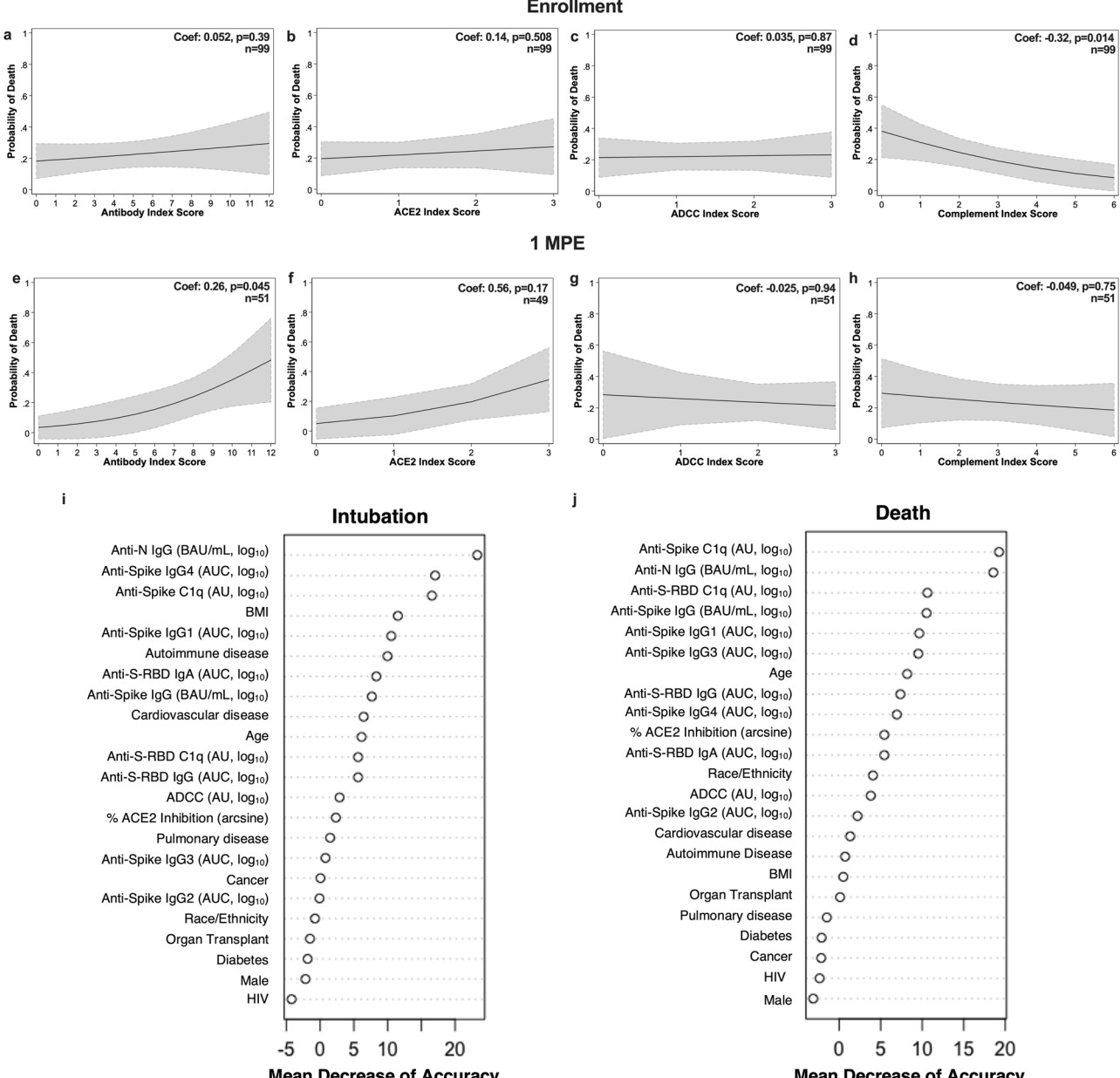

**Fig. 7 | Logistic regression of COVID-19 death by indexed antibody variables among hospitalized COVID-19 patients at enrollment and 1-month post-enrollment (MPE).** Logistic regression modeling for death among hospitalized COVID-19 patients by indexed scores based on quartiles of (**a**, **e**) binding, (**b**, **f**) ACE2 inhibition, (**c**, **g**) ADCC, or (**d**, **h**) complement fixation at enrollment or 1 MPE, respectively. Predicted probabilities from logistic regression models are graphed in black with 95% confidence intervals shaded in gray. **i**, **j** Random Forest variable importance plots were used to determine the relative ranking of different demographic and serological variables in descending order of importance, expressed as mean decrease accuracy, for the classification of intubation or death among hospitalized patients at enrollment. Exclusion of serological variables from models, particularly those >10% mean decrease accuracy, would result in reduced model accuracy for classifying patients as intubated or deceased.

isotypes, virus-neutralizing responses, and non-neutralizing activities, against multiple SARS-CoV-2 epitopes to provide a deep interrogation of the antibody landscape in a cohort of COVID-19 patients. Using machine learning algorithms, we identified the characteristics of the antibody landscape that could predict whether a patient would succumb to or recover from COVID-19.

Systemic complement activation and the ability of anti-S antibodies to induce ADCC were determinants of COVID-19 severity[38–41]. In our cohort, hospitalized patients consistently had higher levels of antibody-mediated complement activation, as measured by C1q of the classical pathway, compared to non-hospitalized patients. Among the hospitalized patients,

anti-S and anti-S-RBD antibody-mediated complement activation was lower in those who died compared to those who recovered from COVID-19. We expected that Fc-mediated antibody functions would increase like anti-S and anti-S-RBD antibody titers among patients hospitalized with more severe COVID-19. In contrast, among patients with progressively worsening disease, antibodies to SARS-CoV-2 had a reduced capacity to activate complement and ADCC, which could contribute to a reduced ability to clear the virus. The role of complement activation in COVID-19 disease severity remains understudied. Recently, Cervia-Hasler, et al.[6] reported that severely ill patients had similar levels of C7 complexes compared to mildly ill patients during active acute COVID-19, but among patients with active long COVID

at the 6-month follow-up levels of C7 and complement activity were elevated, suggesting complement dysregulation may be associated with severity and persistence of disease by impacting coagulation and tissue damage. These findings highlight the need to better understand the non-neutralizing functions of antibodies to SARS-CoV-2 during COVID-19, their predictive value for disease outcomes, and the mechanisms of functional heterogeneity.

Other studies have highlighted the importance of antibody biomarkers in defining the COVID-19 outcome although the results are inconsistent, likely due to differences in study design, patients' characteristics (e.g., age, sex, ethnicity, etc.), antibody assays, and analytical methods (e.g., the groups with which comparisons are made). For example, ref. 42 showed an association of IgG antibody titer at the time of hospital admission with the requirement of mechanical ventilation, while ref. 43 showed that antibody responses are not significantly different between discharged and deceased COVID-19 patients, except for antibodies towards disordered linker region of N protein. De Vito et al. [44] used multivariate Cox regression modeling to show that anti-N IgG titers at hospital admission are independently associated with the risk of death from COVID-19. Smit et al[45]. showed a lower virus-neutralizing antibody titer during hospital admission in fatal versus non-fatal cases of COVID-19, while ref. 13 showed that reduced neutralization potency, but not neutralizing antibody titers, is associated with death from COVID-19. Our data support and expand on previous studies by illustrating that elevated binding and virus neutralization and lower levels of complement-fixing antibody, together with elevated cytokine and chemokine responses during enrollment, are associated with the likelihood of death and intubation from COVID-19 among hospitalized patients.

In our study, the application of machine learning algorithms, such as random forest models, allowed for the identification of the variables, including sociodemographic and immunological measures, that were most predictive of severe COVID-19 outcomes (i.e., intubation or not; deceased or not) in our dataset. Machine learning has been applied to -omic datasets and infectious disease studies with sociodemographic and clinical variables (e.g., age, sex, comorbidities, medications, vital signs, symptoms, lab tests, etc.); however, machine learning has been underutilized with immunological datasets[46–51]. In our study, upper respiratory tract viral RNA levels at the time of enrollment and sociodemographic factors, such as age, sex, or BMI, were not strongly predictive of intubation or death from COVID-19 relative to the serological variables in this dataset. Using enrollment data, our machine learning models revealed that the strongest predictors of subsequent intubation or death from COVID-19 among hospitalized patients were elevated IgG binding antibodies that recognize SARS-CoV-2 N and S proteins and virus-specific antibodies that activate complement.

This study has limitations. We used samples of convenience collected during the pandemic with the non-hospitalized and hospitalized cohorts from two separate parent studies. As a result, the enrollment timepoints were partially asynchronous and the days post-PCR test or enrollment may confound the comparisons between the two cohorts. Additionally, our statistical modeling approaches and interpretations from this study may be influenced by the size of the datasets and the specific variables included. Independent validation datasets were not used to assess random forest model generalizability due to the limited sample sizes and lack of externally available datasets with similar variables and units of measurement. The random forest models were intended to provide insight into biological associations of disease rather than to be used as clinical diagnostic tools. Future studies utilizing machine learning algorithms should carefully consider the context, performance metrics, and generalizability of their models, particularly those with clinical applications. Collection of demographic and clinical data (e.g., comorbidities) were conducted with separate surveys in these two parent studies and may differ in how comorbidities were defined.

Immunological datasets are often highly complex with diverse dependent measures across many sample types. The standard practice among researchers has been to perform over-simplified analyses, such as parametric or non-parametric pairwise comparisons and regression analyses, that may either limit the ability to identify critical associations or over-

interpret them. Machine learning and artificial intelligence (AI) methods now offer unique perspectives for interrogating data with a systems-level approach. Machine learning and AI can inform diagnoses, outcomes, therapeutic targets, and immune profiles for a wide range of diseases with considerable applications in biomedicine and immunological research. We showed the application of machine learning for identifying serological biomarkers at hospitalization enrollment that may predict outcomes of critically ill COVID-19 patients, which has application for elucidating correlates of disease severity as the COVID-19 epidemic continues or for future pandemic preparedness.

## Data availability
All anonymized data that support the findings of this study and source data for the figures are available through the NIH/NCI Serological Sciences Network for COVID-19 (SeroNet) data repository, accessible via ImmPort (https://www.immport.org) under study accession SDY2511. All other data are available from the corresponding author upon reasonable request.

## Code availability
The authors did not develop new software or use custom code in this study. Code for the random forest models can be accessed at https://github.com/ayin0510/JH-EPICS. RStudio 2022.07.02.576, Stata 17.0, and GraphPad Prism were used for analyses.

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

## Acknowledgements

This work is conducted as part of the Johns Hopkins Excellence in Pathogenesis and Immunity Center for SARS-CoV-2 (JH-EPICS) project, which is supported by funding from the National Institute of Health (NIH)/ National Cancer Institute (NCI) grant U54CA260492 (SLK and ALC). Funding also was received from the Sherrilyn and Ken Fisher Center for Environmental Infectious Diseases at the Johns Hopkins University School of Medicine (SD). We would like to thank the study participants who donated their plasma and the clinical staff at Johns Hopkins Healthcare System. The specimens utilized for this publication were part of the Johns Hopkins

Biospecimen Repository, which is based on the contribution of many patients, research teams, and clinicians.

## Author contributions
S.D., A.P., F.A., S.L.Z., and S.L.K. conceptualized the experimental design and group comparisons for this study. P.S., Z.O.D., J.P.H., A.A.R.A., Y.C.M., and A.L.C. identified, acquired, stored, and distributed patient samples and clinical and demographic data for analyses. S.D., M.E-.S., N.P., T.S.J., M.I.T-.Z., K.K., J.S.L., J.R.S., H.-S.P., M.A.P., C.C., A.G., S.K.M., C.D.H., and A.H.K. processed samples and completed technical assays. S.J.G. and M.J.B. provided necessary reagents. A.Y. and S.L.Z. conducted statistical analyses and developed predictive models. S.D., A.Y., M.E-.S., and S.L.K. developed all tables and figures. S.D., A.Y., and S.L.K. wrote the initial draft of the manuscript. All authors provided substantive edits and approved the final draft of the manuscript.

## Competing interests
The authors declare no competing interests.

## Additional information

[1]W. Harry Feinstone Department of Molecular Microbiology and Immunology, Johns Hopkins Bloomberg School of Public Health, Baltimore, MD, USA. [2]Division of Rheumatology, Johns Hopkins School of Medicine, Baltimore, MD, USA. [3]Department of Medicine, Johns Hopkins School of Medicine, Baltimore, MD, USA. [4]Department of Environmental Health and Engineering, Johns Hopkins Bloomberg School of Public Health, Baltimore, MD, USA. [5]Division of Infectious Diseases, Department of Medicine, Johns Hopkins School of Medicine, Baltimore, MD, USA. [6]Department of Biological Chemistry, Johns Hopkins School of Medicine, Baltimore, MD, USA. [7]Department of Chemical and Biomolecular Engineering, Advanced Mammalian Biomanufacturing Innovation Center, Johns Hopkins University, Baltimore, MD, USA. [8]Department of Epidemiology, Johns Hopkins Bloomberg School of Public Health, Baltimore, MD, USA. [9]Department of International Health, Johns Hopkins Bloomberg School of Public Health, Baltimore, MD, USA. [10]Department of Biostatistics, Johns Hopkins Bloomberg School of Public Health, Baltimore, MD, USA. [11]Present address: Department of Diagnostic Medicine/Pathobiology, College of Veterinary Medicine, Kansas State University, Manhattan, KS, USA. [12]These authors contributed equally: Santosh Dhakal, Anna Yin, Marta Escarra-Senmarti. ✉e-mail: sklein2@jhu.edu

