## [Peer review file · Communications Medicine]

Reviewers' comments:

Reviewer #1 (Remarks to the Author):

The authors described identifying serological predictors for COVID-19 severity and outcomes using machine learning models. They have analyzed 178 COVID-19 patients with different severity and outcomes (hospitalized vs non-hospitalized; WHO severity scores from 1-8) for their virus RNA, cytokines/chemokines, binding antibodies, ACE2 binding inhibition, and Fc effector antibody response against SARS-CoV-2 in both mucosal and plasma at enrollment and subsequent time points. The goal of the study was to determine whether serological factors could aid in predicting the clinical outcome of the infection. Cytokine and chemokine levels were associated with higher severity among hospitalized patients, whereas mucosal antibodies did not show a difference. As previously reported, higher plasma antibodies were observed among those hospitalized, including ACE inhibition assays. They identified that IgG binding and ACE2 binding inhibition were associated with more severe disease, whereas, anti-Spike antibody-mediated complement activation was reduced among those with severe outcome. Application of random forest to other demographic data revealed an association of severe outcome with intubation and death.

While primarily based on convenience samples, this is a well-considered and carefully performed study. The data are well described and easily assessed in the figures. The collection of longitudinal samples and measures of antibody levels over time, as well as other key lab outcomes, provide interesting observational data on the COVID-19 disease course.

As the authors describe, there is not a lot of new information emerging here. Higher antibody titers in more severe disease has already been reported. Moreover, this finding was more apparent later in the disease course (1 month after enrollment), and thus less likely to help with initial triage and management.

The most important question is whether early measurements of Ig levels, cytokines, chemokines, ADCC, etc. can predict disease course. Late measurements are interesting but less actionable and will add less to patient assessment.

The data here suggest that the differences were more apparent later in the disease course and that there was significant overlap among cases early in the disease.

It would be useful to see what ROC statistics look like for predicting outcome based on early (i.e., enrollment) antibody (and other) measurements.

Similarly, the usefulness of the AI models is unclear, given that intubation and death are extreme outcomes and may have included data from the later samples.

This is a good study from an observational standpoint and its embedded information will be very helpful for future understanding of COVID-19 clinical course. It is unclear what it specifically adds to our ability to predict COVID-19 outcome.

Reviewer #2 (Remarks to the Author):

The manuscript titled "Application of machine learning models to identify serological predictors of COVID-19 severity and outcomes" aims to uncover relevant patterns in COVID-19 patients' antibody responses. The approach uses machine learning techniques, which is commendable for its innovation. However, this review highlights major concerns that require attention to solidify the reliability and interpretability of the results.

Machine Learning application:

Regarding ML application, the authors seem to have used the Random Forest method correctly for multivariable classification problems. However, the transparency of model performance is critical for trustworthiness and reproducibility, which appears to be an area lacking thoroughness. The authors report AUC values for their models, which provides a summary of model performance. However, these values alone don't offer a complete picture of model performance; specificity, sensitivity, and precision metrics would provide a more rounded understanding.

Moreover, the use of the complete dataset for model training risks overfitting. Ideally, a separate independent test set should have been withheld for final model validation.

Out-Of-Bag (OOB) error estimation, while a valuable internal evaluation metrics within Random Forest, should not replace this important step.

The lack of critical details such as the number of individuals within each training and test set, and the distribution of those that die vs are intubated, creates uncertainty around

model validation. Reporting these metrics would greatly enhance the credibility and transparency of the models.

Additionally, although authors provide a ranking of variable importance, there seems to be no discussion about potential confounding variables in their analysis.

An interpretation of these models' findings within the context of potential confounding factors would be beneficial.

In summary, while Random Forest appears to be correctly applied for the stated objective, there is room for improvement in the methods of model performance evaluation and reporting for a more rigorous, transparent, and reproducible study.

Patient Comparison:

The authors' approach to comparing the non-hospitalized and hospitalized groups based on sample collection timings post-enrollment could present potential challenges for accurate interpretation. The main issue lies in confounding by differing disease timelines for these two patient groups. Hospitalized individuals are typically in the earlier phase of infection, when the immune response is more active, resulting in elevated antibody levels. Non-hospitalized individuals, sampled anywhere from 18 to 91 days post-PCR confirmation, could be in the recovery phase, where antibody levels have been observed to decline. This asynchronous comparison implies that differences in observed antibody levels may not be due to disease severity but because of the timing of sample collection post-infection, which is not controlled for in this study design.

A spread of 18-91 days post-PCR confirmation in the non-hospitalized group engulfs different disease phases where immune responses can significantly vary. Hence, grouping all these individuals as '1-month post-enrollment' introduces substantial variance in the dataset.

The 3-month post-infection timepoint is likely to show substantially lower antibody responses due to the natural waning of the immune response, easily confounding the analysis. The conclusion that hospitalized individuals have higher antibody levels has a risk of circular reasoning, as these samples come from an earlier and acute phase of infection, inherently showing higher antibody levels. It might not necessarily indicate intrinsic immunological differences based on disease severity alone.

In epidemiological studies, the choice of comparison groups and controlling for temporal factors is of utmost importance to avoid biased conclusions. In this case, the analysis does not control for temporal differences, which can have a significant effect on the measured biomarkers.

In addition to the two major concerns described above, there is also critical usage of terminology related to machine learning.

First, the title of the manuscript requires rectification. The usage of "machine learning models" in the title is misleading, as it is the machine learning algorithms that generate these models. The current title version infers that authors used already published ML models which they didn't. Hence, the title should be corrected to reflect the use of "machine learning algorithms" to ensure a precise representation of your research's methodology.

Secondly, the phrase "automated intelligence" used as a keyword is incorrect. This term is not used in scientific literature and might confuse readers. "Automated intelligence" is a misnomer and should be replaced with "machine learning" or "artificial intelligence."

The identified critical issues should be properly addressed. It's fundamental to correct the misnomer in the title and appropriately define the keywords to ensure an accurate representation of the research. Moreover, a thorough revision that responds to the previously noted methodological and comparative concerns is of utmost importance. These changes are crucial in enhancing the manuscript's clarity, credibility, and overall contribution to the field.

We would like to thank the reviewers for their thorough review of our manuscript. Their critique has helped to strengthen this paper. Responses are documented below in blue.

Reviewers' comments:

Reviewer #1 (Remarks to the Author):

The authors described identifying serological predictors for COVID-19 severity and outcomes using machine learning models. They have analyzed 178 COVID-19 patients with different severity and outcomes (hospitalized vs non-hospitalized; WHO severity scores from 1-8) for their virus RNA, cytokines/chemokines, binding antibodies, ACE2 binding inhibition, and Fc effector antibody response against SARS-CoV-2 in both mucosal and plasma at enrollment and subsequent time points. The goal of the study was to determine whether serological factors could aid in predicting the clinical outcome of the infection. Cytokine and chemokine levels were associated with higher severity among hospitalized patients, whereas mucosal antibodies did not show a difference. As previously reported, higher plasma antibodies were observed among those hospitalized, including ACE inhibition assays. They identified that IgG binding and ACE2 binding inhibition were associated with more severe disease, whereas anti-Spike antibody-mediated complement activation was reduced among those with severe outcome. Application of random forest to other demographic data revealed an association of severe outcome with intubation and death.

While primarily based on convenience samples, this is a well-considered and carefully performed study. The data are well described and easily assessed in the figures. The collection of longitudinal samples and measures of antibody levels over time, as well as other key lab outcomes, provide interesting observational data on the COVID-19 disease course.

As the authors describe, there is not a lot of new information emerging here. Higher antibody titers in more severe disease has already been reported. Moreover, this finding was more apparent later in the disease course (1 month after enrollment), and thus less likely to help with initial triage and management.

The most important question is whether early measurements of Ig levels, cytokines, chemokines, ADCC, etc. can predict disease course. Late measurements are interesting but less actionable and will add less to patient assessment.

Response: We thank the reviewer for their excellent suggestions. We agree that later serological measurements may be less actionable than earlier measurements. To clarify, we used serological and sociodemographic data from enrollment (early measurement) in our random forest models to predict subsequent COVID-19 hospitalization outcome. These models did not include serological or sociodemographic data from later timepoints (e.g., 1 MPE). This is now clarified throughout the Abstract (page 2, line 51), Introduction (page 4, line 82), Methods (page 5, line 98), Results (page 18-19) and Discussion (page 22, line 493; page 23, lines 516-517) sections.

The data here suggest that the differences were more apparent later in the disease course and that there was significant overlap among cases early in the disease.

It would be useful to see what ROC statistics look like for predicting outcome based on early (i.e., enrollment) antibody (and other) measurements.

Response: Thank you for suggesting the inclusion of ROC statistics for the random forest models using enrollment data. We have included additional RF model performance details in supplementary figures 9-11 to substantiate the use of RF models, including AUROC, OOB error rates, sensitivity, and specificity values. These data are further incorporated into the Results (on page 18, lines 403-410; 413-416; page 19, lines 427-433).

Similarly, the usefulness of the AI models is unclear, given that intubation and death are extreme outcomes and may have included data from the later samples.

Response: Thank you for pointing out the need for clarification of the usefulness of AI/machine learning in this study. First, our random forest models included data from enrollment (i.e., early measurements), and not later timepoints, to assess how well these variables can predict subsequent disease outcomes among hospitalized COVID-19 patients. Secondly, while intubation and death due to COVID-19 are extreme outcomes that may not be as prevalent now, they are still of mechanistic interest and have immunological significance. Understanding the relationship between these serological measures with extreme disease outcomes may have applications and relevancy for future pandemics such that serological and sociodemographic profiles of patients can be utilized to predict subsequent disease outcomes upon hospitalization enrollment. Finally, the implications of these early serological measures on long COVID suggests additional relevance of these finds, which is highlighted in the Introduction (on page 3, lines 60-61).

This is a good study from an observational standpoint and its embedded information will be very helpful for future understanding of COVID-19 clinical course. It is unclear what it specifically adds to our ability to predict COVID-19 outcome.

Response: Thank you for bringing this to our attention. We have clarified the wording to emphasize the applications and utility of these models for predicting disease outcomes. These models suggest that serological measures can better predict disease outcomes than sociodemographic variables, providing mechanistic and clinical insight, that have applications beyond COVID-19 for future pandemics as well.

Reviewer #2 (Remarks to the Author):

The manuscript titled "Application of machine learning models to identify serological predictors of COVID-19 severity and outcomes" aims to uncover relevant patterns in COVID-19 patients' antibody responses. The approach uses machine learning techniques, which is commendable for its innovation. However, this review highlights major concerns that require attention to solidify the reliability and interpretability of the results.

Machine Learning application:

Regarding ML application, the authors seem to have used the Random Forest method correctly for multivariable classification problems. However, the transparency of model performance is critical for trustworthiness and reproducibility, which appears to be an area lacking thoroughness. The authors report AUC values for their models, which provides a summary of model performance. However, these values alone don't offer a complete picture of model performance; specificity, sensitivity, and precision metrics would provide a more rounded understanding.

Moreover, the use of the complete dataset for model training risks overfitting. Ideally, a separate independent test set should have been withheld for final model validation.

Out-Of-Bag (OOB) error estimation, while a valuable internal evaluation metrics within Random Forest, should not replace this important step.

Response: We thank the reviewer and greatly appreciate their suggestion for additional model metrics to better support the use of these models. Out-of-bag (OOB) evaluation metrics allow us to evaluate the performance of the RF models with one training dataset where the OOB error estimations are calculated based on the data that were not used to train the model after bootstrap resampling. Alternatively, we have also included evaluation metrics after performing stratified 10-fold cross-validation random forest modeling to address the reviewer's concerns about overfitting in supplementary figure 11 and in the results section (page 19, lines 427-433). This approach partitions the dataset into 10 folds such that the model is trained on 9 of the folds (90% of the dataset) and tested on the remaining 1-fold (10% of the dataset), which is then repeated 10 times with each fold serving as a test set once. The methods for the random forest models have been updated on page 11-12, lines 251-270. We present two approaches of assessing random forest model performance with consistent results from both suggesting that serological variables are more useful for predicting COVID-19 hospitalization outcome than sociodemographic variables. Details about these additional model metrics can be found in the Results section on pages 18-19 and supplementary figures 9-11.

The lack of critical details such as the number of individuals within each training and test set, and the distribution of those that die vs are intubated, creates uncertainty around model validation. Reporting these metrics would greatly enhance the credibility and transparency of the models.

Response: We greatly appreciate the reviewer's suggestion and have updated the manuscript to include further details about the individuals used in the random forest modeling datasets. In brief, complete datasets from enrollment, including over 20 serological and sociodemographic variables for 98 hospitalized participants, were used for our random forest modeling. Of the 98 hospitalized patients, 45 (46%) were intubated and 21 (21%) subsequently died due to COVID-19. The parameters for our 10-fold cross-validation with random forest modeling is set so that there is approximately balanced distribution of the outcome variable in each fold. With the 10-fold cross-validation approach, the dataset is then partitioned into 10 folds and the model is trained on 9 of the folds and tested on the remaining fold. With a total of 98 patients in the dataset, the 9 folds represent 90% of the dataset with an average of 88 hospitalized patients and the remaining fold represents 10% of the dataset with an average of 10 hospitalized patients. Modifications to the manuscript can be found in the Methods section on page 11, lines 251-255.

Additionally, although authors provide a ranking of variable importance, there seems to be no discussion about potential confounding variables in their analysis.

Response: Thank you for pointing out the necessity for consideration of confounding variables. Indeed, confounding variables are both statistically and mechanistically important to consider. To clarify, the main goal of machine learning algorithms, such as random forest modeling, is to make accurate predictions for classification whereas statistical analyses focus on measuring the effect of a predictor on an outcome. These variable importance plots, therefore, enable us to evaluate which variables, relative to the others, were best able to classify COVID-19 hospitalization outcomes in our dataset using the random forest models, regardless of confounding status. The ranking of variables is not necessarily indicative of a variable's effect on the outcome.

We used random forest modeling to assess how useful serological and sociodemographic variables are for classifying COVID-19 hospitalization outcomes. We also demonstrated that random forest models using only sociodemographic variables (e.g., age, BMI, etc.), potential

confounders, to classify hospitalization have much lower accuracy than models using only serological variables. This has important applications beyond COVID-19 for future pandemics such that hospitalization outcome may be anticipated upon hospital enrollment of the patient based on serological and sociodemographic profiles.

Partial dependence plots, which are useful for examining how the changes in one variable, controlling for all others, impact the random forest model's predictions, are included below. Likewise, bivariate dependence plots demonstrate how changing the relationship of two variables, while controlling for all others, impacts the model's predictions. These plots have been included to address concerns about confounders and provides a deeper understanding of the relationship of the variables with the classification of COVID-19 hospitalization outcomes. Details about dependence plots have been included in the supplementary figures 9-10.

An interpretation of these models' findings within the context of potential confounding factors would be beneficial.

Response: We thank the reviewer for their valuable feedback. We considered potential confounding factors, such as patient age and sex, in our study and analyses. Regarding the comparisons made between hospitalized and non-hospitalized patients at 1 MPE, sex and age can be considered confounders of hospitalization status and serological measures. Using linear regression analysis, controlling for sex and age, we found that hospitalized patients consistently had higher serological measures than non-hospitalized patients at 1 MPE, which has been updated in figure 3 and in the methods on page 11-12, lines 234-235.

Among the hospitalized patients, we did not identify an association of sex and age with severity of disease as the distributions of sex and age were similar across moderate, severe, and deceased patients. Random forest modeling was used to examine the predictive power of the variables relative to each other. As such, we needed to be able to use all variables, regardless of confounding status, to determine which variables contributed the most predictive accuracy of our models. In our study, serological variables consistently had superior predictive ability for hospitalization outcomes relative to sociodemographic variables.

To better address the reviewer's concerns, we have included partial dependence and bivariate dependence plots in supplementary figures 9-10 with details on page 18, lines 403-410 and 413-416. These plots demonstrate how changing one or two of the most predictive variables, while controlling for all others, impacts the predicted probability of intubation or death.

In summary, while Random Forest appears to be correctly applied for the stated objective, there is room for improvement in the methods of model performance evaluation and reporting for a more rigorous, transparent, and reproducible study.

Patient Comparison:

The authors' approach to comparing the non-hospitalized and hospitalized groups based on sample collection timings post-enrollment could present potential challenges for accurate interpretation. The main issue lies in confounding by differing disease timelines for these two patient groups. Hospitalized individuals are typically in the earlier phase of infection, when the immune response is more active, resulting in elevated antibody levels. Non-hospitalized individuals, sampled anywhere from 18 to 91 days post-PCR confirmation, could be in the recovery phase, where antibody levels have been observed to decline. This asynchronous comparison implies that differences in observed antibody levels may not be due to disease

severity but because of the timing of sample collection post-infection, which is not controlled for in this study design.

Response: Thank you for pointing this out, this has been a major challenge in our study. Indeed, the potential confounding by the days post-PCR confirmation or enrollment was seriously considered prior to analyses. As these are samples of convenience collected through two independent studies and cohorts, the studies were not designed to have synchronized sample collection timepoints. The non-hospitalized patients were enrolled when they arrived at Johns Hopkins for PCR-testing and samples were collected with highly variable days after enrollment. The hospitalized patients were enrolled when they were admitted into the hospital with COVID-19 and had their initial blood sample collected. As a result, the “baseline” for these two cohorts is inherently different as the reviewer suggested. While we can theoretically control for the days post-enrollment in our analyses, it may also introduce additional confounding that we simply cannot adequately control for given data we have and the study designs.

Asynchronous comparison between non-hospitalized and hospitalized patients is not ideal; however, we compared non-hospitalized and hospitalized to demonstrate that more severely ill patients had higher antibody levels than non-hospitalized patients, consistent with the current literature. The primary focus of our paper was not the comparisons between non-hospitalized and hospitalized patients, but rather the comparisons among hospitalized patients. To address the reviewer’s concern, we would like to point them to supplementary figure 1. We demonstrated that the antibody levels were similar over days post-enrollment among the non-hospitalized given the timeframe of samples collected based on spearman correlation analysis. We used this as justification for restricting non-hospitalized patient samples between 31 and 61 days after enrollment to be grouped for 1 MPE to compare against hospitalized patients. This revised information can be found in the methods on pages 4-5, lines 94-107.

A spread of 18-91 days post-PCR confirmation in the non-hospitalized group engulfs different disease phases where immune responses can significantly vary. Hence, grouping all these individuals as '1-month post-enrollment' introduces substantial variance in the dataset.

Thank you for pointing this out. To clarify, we did not use non-hospitalized data spanning 18-91 days post-PCR confirmation in our 1 MPE analyses to compare against hospitalized. We restricted the non-hospitalized samples from 31-61 days post-PCR confirmation to compare against hospitalized samples at 1 MPE, which was calculated based on the mean and standard deviation of days post-PCR confirmation. We have clarified the Methods (pages 4-5, lines 94-107) to more accurately state how we grouped non-hospitalized data for 1 MPE analyses.

The 3-month post-infection timepoint is likely to show substantially lower antibody responses due to the natural waning of the immune response, easily confounding the analysis. The conclusion that hospitalized individuals have higher antibody levels has a risk of circular reasoning, as these samples come from an earlier and acute phase of infection, inherently showing higher antibody levels. It might not necessarily indicate intrinsic immunological differences based on disease severity alone.

In epidemiological studies, the choice of comparison groups and controlling for temporal factors is of utmost importance to avoid biased conclusions. In this case, the analysis does not control for temporal differences, which can have a significant effect on the measured biomarkers.

Response: We thank the reviewer for their suggestion and agree that temporal changes are important considerations any study. As mentioned previously, our non-hospitalized cohort did indeed have more variability in the timing of sample collection. In the supplementary figure 1, we showed that the temporal variability of sample collection did not have significant correlation with

days post-PCR confirmation for the non-hospitalized patients. Sample collection for hospitalized patients at enrollment and 1 MPE were more organized with less variability in timing of sample collection. In the longitudinal analyses of hospitalized patients, we include continuous days post-enrollment as a variable to examine the effect of time on serological measures and, therefore, cannot control for time in these models. Lastly, our random forest models only used data from the hospitalized patients at enrollment and did not include data from other timepoints. Modifications in the manuscript are contained in the methods (pages 4-5, lines 94-107), results (page 18, line 405; page 19, line 419), and discussion (page 22, lines 493; page 23, line 516-517).

In addition to the two major concerns described above, there is also critical usage of terminology related to machine learning.

First, the title of the manuscript requires rectification. The usage of "machine learning models" in the title is misleading, as it is the machine learning algorithms that generate these models. The current title version infers that authors used already published ML models which they didn't. Hence, the title should be corrected to reflect the use of "machine learning algorithms" to ensure a precise representation of your research's methodology.

Secondly, the phrase "automated intelligence" used as a keyword is incorrect. This term is not used in scientific literature and might confuse readers. "Automated intelligence" is a misnomer and should be replaced with "machine learning" or "artificial intelligence."

Response: We greatly appreciate the reviewer's suggestions. We have revised the title and use of "artificial intelligence" in our manuscript.

The identified critical issues should be properly addressed. It's fundamental to correct the misnomer in the title and appropriately define the keywords to ensure an accurate representation of the research. Moreover, a thorough revision that responds to the previously noted methodological and comparative concerns is of utmost importance. These changes are crucial in enhancing the manuscript's clarity, credibility, and overall contribution to the field.

Reviewers' comments:

Reviewer #2 (Remarks to the Author):

Upon careful review of the manuscript, I must highlight two major problems that significantly impact the credibility and utility of the presented models. Addressing these issues is crucial for the advancement of predictive modeling in healthcare settings.

1. Lack of Independent Evaluation: The manuscript does not document the use of an independent test set for model evaluation. An independent test set, ideally segregated from the data before any training or tuning activities commence, is fundamental for assessing a model's performance. This independent evaluation is crucial to confirm the model's ability to generalize to completely unseen data. The absence of such a validation step raises concerns about the model's reliability and its potential overfitting to the training dataset, thus questioning its applicability in real-world clinical settings.

2. Lack of Thorough Evaluation of Models: Specifically, the evaluation of the model predicting patient mortality raises significant concerns. The reliance on AUROC (Area Under the Receiver Operating Characteristic Curve) as the primary metric for model performance, particularly in the context of highly imbalanced data (77 survived and 21 died), is misleading. While an AUROC of 0.7 might suggest moderate discriminative ability, a deeper analysis reveals a disturbingly high false negative rate. The model's sensitivity, or its ability to correctly identify patients who died, is alarmingly low at approximately 5%. This means the model correctly predicts the mortality of only about 1 out of the 21 patients who actually died. In a healthcare setting, where the cost of missing at-risk patients can be exceedingly high, this oversight is unacceptable. The model's failure to identify patients at risk of death could lead to inadequate patient care and missed opportunities for crucial interventions. The manuscript does not adequately address or discuss this critical limitation.

These problems collectively undermine the manuscript's contribution to predictive modeling in healthcare. The lack of an independent test set evaluation compromises the model's demonstrated ability to perform in real-world scenarios. Simultaneously, the misinterpretation and inadequate evaluation of the model predicting patient death, particularly its neglect to consider the implications of a high false negative rate in a healthcare context, is a significant oversight.

To enhance the manuscript and the models it presents, I strongly recommend the following actions:

- Incorporate an independent test set that has not been used in any capacity during the

model's development for a more reliable assessment of its generalizability.

- Re-evaluate the model predicting patient mortality with a focus on metrics that are more informative for imbalanced datasets, such as sensitivity, precision-recall curves, or the F1 score. This re-evaluation should include a thorough discussion of the model's performance limitations, especially its ability to identify patients at high risk of mortality.

Addressing these issues is essential for presenting a more accurate and clinically applicable predictive model. The potential for predictive modeling to improve patient outcomes is immense, but only if such models are developed, evaluated, and presented with rigor and transparency.

Upon careful review of the manuscript, I must highlight two major problems that significantly impact the credibility and utility of the presented models. Addressing these issues is crucial for the advancement of predictive modeling in healthcare settings.

1. Lack of Independent Evaluation: The manuscript does not document the use of an independent test set for model evaluation. An independent test set, ideally segregated from the data before any training or tuning activities commence, is fundamental for assessing a model's performance. This independent evaluation is crucial to confirm the model's ability to generalize to completely unseen data. The absence of such a validation step raises concerns about the model's reliability and its potential overfitting to the training dataset, thus questioning its applicability in real-world clinical settings.

We thank the reviewer for their feedback and agree that consideration for applicability in a real-world clinical setting is critical for models that are intended for clinical applications. Independent datasets for hospitalized COVID-19 patients which include the same serological, demographic, and clinical variables as our dataset for validating the model's performance are difficult to obtain, particularly with the extent or depth of measures of our dataset, making these independent evaluations difficult. In an effort to address the reviewer's concern, we obtained a similar dataset from another research group to integrate with our existing data that would enable us to partition out an independent test set. However, the data were not sufficiently comparable due to discrepancies in variables and units of measurements; thus, it could not be integrated with our larger dataset. For this independent dataset alone, the random forest model for death had an AUROC=0.55, meaning that the model's classification of patients that died was only slightly better than random chance. Our struggles in obtaining independent datasets for validation and to assess model generalizability underscore the current challenge of using machine learning in this field—serological datasets are too varied in measures and units across studies. In our case, we sacrificed sample size for a greater depth of measurements, including measures of non-neutralizing functions that have not been widely evaluated among severe COVID-19 patients. Our hope is that this study can contribute to the publicly available datasets that can be used in future studies as an independent test set as the reviewer has suggested.

Given our limited data availability and sample sizes, we are not proposing that our models be used for clinical applications. Rather, we are using machine learning to provide etiologic insight for possible biomarkers of critically ill hospitalized COVID-19 patients at enrollment. In our manuscript, we included results from two random forest evaluation approaches—one by OOB error rate, another by stratified k-fold cross-validation. With the out-of-bag random forest approach, the algorithm randomly samples a subset of the data for training and using the remaining data for testing. With the k-fold cross-validation approach, the algorithm partitions the data into 90% for training and 10% for testing, repeating this 10 times. These methods eliminate the need for independent evaluation of the model on another set of unseen data. We have specified the lack of an independent evaluation, due to limited data, as a limitation on page 23, lines 518-524.

2. Lack of Thorough Evaluation of Models: Specifically, the evaluation of the model predicting patient mortality raises significant concerns. The reliance on AUROC (Area Under the Receiver Operating Characteristic Curve) as the primary metric for model performance, particularly in the context of highly imbalanced data (77 survived and 21 died), is misleading. While an AUROC of 0.7 might suggest moderate discriminative ability, a deeper analysis reveals a disturbingly high false negative rate. The model's sensitivity, or its ability to correctly identify patients who died, is alarmingly low at approximately 5%. This means the model correctly predicts the mortality of only about 1 out of the 21 patients who actually died. In a healthcare setting, where the cost of

missing at-risk patients can be exceedingly high, this oversight is unacceptable. The model's failure to identify patients at risk of death could lead to inadequate patient care and missed opportunities for crucial interventions. The manuscript does not adequately address or discuss this critical limitation.

These problems collectively undermine the manuscript's contribution to predictive modeling in healthcare. The lack of an independent test set evaluation compromises the model's demonstrated ability to perform in real-world scenarios. Simultaneously, the misinterpretation and inadequate evaluation of the model predicting patient death, particularly its neglect to consider the implications of a high false negative rate in a healthcare context, is a significant oversight.

To enhance the manuscript and the models it presents, I strongly recommend the following actions:

- Incorporate an independent test set that has not been used in any capacity during the model's development for a more reliable assessment of its generalizability.
- Re-evaluate the model predicting patient mortality with a focus on metrics that are more informative for imbalanced datasets, such as sensitivity, precision-recall curves, or the F1 score. This re-evaluation should include a thorough discussion of the model's performance limitations, especially its ability to identify patients at high risk of mortality. Addressing these issues is essential for presenting a more accurate and clinically applicable predictive model. The potential for predictive modeling to improve patient outcomes is immense, but only if such models are developed, evaluated, and presented with rigor and transparency.

Thank you for the suggestions. We are using machine learning (i.e., random forest modeling) to gain etiologic insights rather than as a diagnostic tool in clinical settings. Indeed, any model that has potential to be used as a clinical diagnostic tool should be developed, tested, and evaluated with rigor. We present a novel application of machine learning that suggests that serological measures, rather than demographics or clinical comorbidities, are associated with disease outcomes for critically ill COVID-19 hospitalized patients. We have revised the wording on page 24, lines 537-540 to clarify this point. With the sparsity of large COVID-19 immunological datasets available for thorough testing and validation, particularly with this level of depth of measures, machine learning-based tools for human diagnoses may not be ready for clinical applications at this time.

For our random forest models, the performance metrics initially reported in our manuscript were for a default probabilistic cutoff of 0.5, which had a lower sensitivity. In the context of random forest modeling, the cutoff represents the probability used to classify a positive or negative class based on the model's predictions. A cutoff of 0.5 would be most useful for balanced datasets; however, in our dataset, there is an imbalance of deaths vs non-deaths with $n=21/98$ deaths (21.4%) as noted by the reviewer. The cutoff can be set at different values, depending on the context of its applications. In some contexts, like clinical diagnoses, higher sensitivity may be more valuable at the cost of lower specificity or vice versa. For our models, we set the cutoff so that the sensitivity and specificity were relatively balanced as our aim was to determine which measures (serological or sociodemographic) would be most predictive of subsequent COVID-19 hospitalization outcome. We have updated the random forest death model performance metrics to be based on a cutoff of 0.21 and report increased model sensitivity (supplemental figure 10). We also have updated the cutoffs for the random forest intubation models to be 0.46 ($n=45/98$ intubated; supplemental figure 9). These details have been updated in our methods (page 12, lines 277-279), limitations (page 23, line 518-524), and supplemental figures 9-10.

To address the reviewer's concern about other metrics of model performance, we have also included the F1 score in the table of each RF model of supplementary figures 9-10. In short, an F1 score closer to 1 represents a model that has higher accuracy with high precision and recall. We find that, regardless of using OOB or k-fold cross-validation approaches, the models including serological measures had better model performance metrics than the models without serological measures, consistent with our findings and conclusion in the manuscript.

Reviewers' comments:

Reviewer #2 (Remarks to the Author):

Dear authors,

Thank you for addressing my previous comments. However, I have identified an additional issue that needs to be addressed.

Since your research involves the development of predictive models, it is essential to share the complete ML models, including the exact datasets used for training. Without these, reproducing your findings is impossible. Given that your entire data analysis relies on ML modeling, it is crucial to share the entire dataset. Statements such as "data available upon request" are not acceptable in this context.

Furthermore, the data uploaded to Immpport is raw and cannot be directly used to validate your findings. A standardized and processed dataset must be made available through repositories like GitHub, Zenodo, or similar, in compliance with NIH data-sharing guidelines and FAIR standards.

Additionally, you did not share any R scripts necessary to perform the exact analysis you conducted, nor did you cite which R packages were used for the analysis. This information is crucial for ensuring the reproducibility and validation of your work.

Without sharing the ML models as R objects, the complete dataset, and the necessary computational code or R scripts and packages, the validity of your findings cannot be reproduced.

Thank you for your attention to this matter.

Reviewer #2 (Remarks to the Author):

Dear authors,

Thank you for addressing my previous comments. However, I have identified an additional issue that needs to be addressed.

Since your research involves the development of predictive models, it is essential to share the complete ML models, including the exact datasets used for training. Without these, reproducing your findings is impossible. Given that your entire data analysis relies on ML modeling, it is crucial to share the entire dataset. Statements such as "data available upon request" are not acceptable in this context.

We appreciate the reviewer's feedback and agree that having the entire dataset publicly available is essential for the reproducibility of findings. The complete dataset (i.e., no missing data) used for the RF models is a subset of the deposited data on ImmPort. If someone is interested in reproducing the RF findings, they can filter the deposited dataset based on the following conditions: hospitalized=1, study timepoint = 0 (enrollment), and not missing any demographic or clinical data. Variables used in the RF models are listed on the variable importance plots and the dataset should be restricted to those measures for classification. Below, we list out the 24 variables used for n=98 samples with complete data in the classification of either death or intubation.

- | | | |
|---------------------------------|------------------------------------|----------------------------------|
| 1. Age | 9. Diabetes | 18. Anti-Spike IgG2 (AUC, log10) |
| 2. Race/ethnicity | 10. Autoimmune disease | 19. Anti-Spike IgG3 (AUC, log10) |
| 3. BMI | 11. Cancer | 20. Anti-Spike IgG4 (AUC, log10) |
| 4. Sex | 12. Death (or intubation) | 21. % ACE2 ancestral, arcsine |
| 5. HIV | 13. Anti-S-RBD IgG (AUC, log10) | 22. Anti-S-RBD C1q (AU, log10) |
| 6. Organ transplant | 14. Anti-S-RBD IgA (AUC, log10) | 23. Anti-Spike C1q (AU, log10) |
| 7. Cardiovascular disease (CVD) | 15. Anti-N IgG (BAU/mL, log10) | 24. ADCC (AU, log10) |
| 8. Pulmonary disease | 16. Anti-Spike IgG (BAU/mL, log10) | |
| | 17. Anti-Spike IgG1 (AUC, log10) | |

Furthermore, we write "All anonymized data that support the findings of this study are available through the NIH/NCI Serological Sciences Network for COVID-19 (SeroNet) data repository, accessible via ImmPort (<https://www.immport.org>) under study accession SDY2511. "All other data are available from the corresponding author upon reasonable request" on page 25, lines 565-569. We specify that all other data, such as dates and detailed clinical measures, may be available upon reasonable request as these data may be subject to further deidentification and/or IRB approvals prior to sharing due to data privacy guidelines (e.g., HIPAA). The deposited data, as is, should be sufficient to replicate the findings.

Furthermore, the data uploaded to Immport is raw and cannot be directly used to validate your findings. A standardized and processed dataset must be made available through repositories like GitHub, Zenodo, or similar, in compliance with NIH data-sharing guidelines and FAIR standards.

We have carefully reviewed the deposited data file and can confirm that the data uploaded to ImmPort contains the log10-transformed serological measures, categorical clinical and demographic variables that were used for our random forest models.

ImmPort is the preferred data repository that NIH SeroNet uses and was required as part of the study funding.

Additionally, you did not share any R scripts necessary to perform the exact analysis you conducted, nor did you cite which R packages were used for the analysis. This information is crucial for ensuring the reproducibility and validation of your work. Without sharing the ML models as R objects, the complete dataset, and the necessary computational code or R scripts and packages, the validity of your findings cannot be reproduced.

We thank the reviewer for pointing this out and we agree that this is necessary for reproducibility. On page 13, lines 282-285 we write “All random forest models were trained using the same seed value to ensure reproducibility. randomForest, caret, ROCR, MLmetrics, and pdp packages in R were used for the random forest modeling, calculating variable importance scores, evaluating performance metrics, and visualizing dependence plots”. We used existing R packages, including randomForest and caret, and their built-in functions to perform random forest modeling. Additional parameters of our RF models are also detailed in the statistical methods section on pages 12-13. To address the reviewer’s concerns, we have also uploaded the R scripts to Github. On page 25, lines 571-572 under the code availability statement, we have included “Code for the random forest models can be accessed at <https://github.com/ayin0510/JH-EPICS>.”

Thank you for your attention to this matter.